# Evolutionary features of microscopic damage in shale under unloading action

**Yanxu Liang[1], Haicheng She[1,2]\*, Wangji Ding[1], Haidong She[3]**

**1** School of Urban Construction, Yangtze University, Jingzhou, China, **2** Key Laboratory of Reservoir and Dam Safety Ministry of Water Resources, Nanjing, China, **3** Datong Hongtai Mine Engineering Construction Co., Ltd. of Jinneng Holding Coal Industry Group, Datong, China

\* shehaicheng@126.com

## Abstract

To reveal the microscopic damage evolution law of rocks under the effect of unloading disturbances with different amplitudes, electron microscope scanning, nuclear magnetic resonance (NMR), and triaxial compression tests were carried out. The evolution patterns of surface and internal pore types and mechanical properties of rock specimens after unloading perturbation were analyzed. In this paper, a classification of the ratio of $d_{max}/d_{min}$ ($d_{max}$ and $d_{min}$ refer to the maximum and minimum pore size of each pore, respectively) is proposed to examine the pore and crack evolution extension development on the surface of the specimen. Meanwhile, the $T_2$ energy spectrum with pore size classification is used to examine the damage quantification law of the pore and crack extension evolution inside the specimen. Finally, the statistical damage model of unloaded disturbed rock is established through theoretical derivation, and the accuracy of the model is verified by experimental data. The study shows that: (1) With the increase of unloading amplitude, there is an increase in the number of nascent cracks and a tendency to expand, which is caused by shear extension cracks; with the increase of unloading amplitude, there is a tendency for the microporosity to shift to the mesoporosity, and the mesoporosity has a tendency to shift to the macroporosity, and there is a decrease in the number of micropores as a whole, which indicates that there is almost no new pore sprouting. (2) When the unloading amplitude is less than 20MPa, with the increase of the unloading amplitude, the pore ratio and expansion rate of the specimen increase slowly; when the unloading amplitude is more than 20MPa, with the increase of the unloading amplitude, the pore ratio and expansion rate of the specimen have a significant tendency to increase. (3) With the increase of unloading amplitude, the shale shear strength limit value decreases more slowly, the modulus of elasticity and shear strength also show a similar pattern of change, and the same way to derive the rock Poisson's ratio does not change much.

**Data availability statement:** All relevant data are within the manuscript and its Supporting information files.

**Funding:** This work was supported by the Natural Science Foundation of Hubei Province (2024AFB878), the Key Laboratory of Reservoir and Dam Safety of the Ministry of Water Resources is open to research funds (YK323003) and Hubei Provincial Key Laboratory of Oil and Gas Drilling and Production Engineering Open Fund (YQZC202204), which are gratefully acknowledged. The funders provided financial support for the purchase of experimental materials, translation and polishing of the manuscript.

**Competing interests:** The authors have declared that no competing interests exist.

## 1 Introduction

With the increasing number of tunnels, pits, slopes, and other projects, the problem of disturbance by various factors has become more and more prominent [1–3]. In these works, the soil and rock bodies are inevitably subjected to unloading disturbances of varying magnitudes, which in turn produce a range of disturbance effects [4]. These disturbance effects will not only affect the stability and safety of the project but also have a certain impact on the surrounding environment [5]. Therefore, the study of the damage characteristics of rocks under the action of unloading perturbation is an important scientific guidance for safety, stability, economy and long-term service performance in engineering.

In recent years, many scholars have done a lot of research on the mechanical properties of rocks under the action of various factors of perturbation, etc., and have achieved rich research results. In terms of impact perturbation: taking red sandstone as the research object, the impact disturbance test was conducted on red sandstone to analyze the mechanical response and deformation mechanism of red sandstone after being subjected to impact disturbance [6], which revealed the damage evolution, critical damage amount, crack extension law and damage mode of red sandstone under repeated impact loads [7]. In terms of unloading disturbance: Fu et al. [8] designed three unloading circumferential pressure tests of sandstone under different initial axial pressures, analyzed the change characteristics of rock porosity and $T_2$ spectral curves by NMR technique, and established the relationship between the degree of damage and the unloading circumferential pressure ratio. Zhou et al. [9] conducted unloaded perimeter pressure tests on rock specimens and NMR tests on unloaded specimens to study the fine damage evolution characteristics of unloaded rocks to obtain the change rules of stress-strain curves, rock porosity, and NMR parameters during the damage process. Shi et al. [10] used true triaxial unloading perturbation tests. Mechanical properties of rocks under different stresses were studied. The crack evolution laws of rocks in different environments were also analyzed. Guo et al. [11] characterized the deformation and fracture patterns of shale samples under different stress paths by triaxial unloading tests under different stress paths. Lei et al. [12] conducted a series of loading and unloading tests on the rock mass around the roadway using high-precision acoustic emission technology. The results of the study provide guidance for deep shaft tunnel support work and disaster prevention and control. Combined with specific engineering examples, the mechanical property changes of soft rock in the unloading process were studied. [13] The effects of unloading rate, initial stress state, and other factors on the mechanical properties of soft rock were analyzed. [14] The deformation, damage mode, and strain energy evolution were studied in conjunction with discrete element simulation, and it was found that the faster the unloading rate, the more violent the damage was. [15] The unloading rate of the soft rock was also analyzed. In terms of hydration disturbance: Pu et al. [16] used SHPB experiments to study the dynamic compressive strength change of sandstone after wet and dry cycles, and used ultrasonic waves to detect the change of wave velocity of sandstone before and after wet and dry cycles, which reflected the change rule of the internal structure of the rock; Dong et al. [17]

carried out a uniaxial compression test on sandstone samples with different submergence heights, and used a high-speed camera and an acoustic emission monitoring system to simultaneously monitoring the damage process, and established a damage evolution model using damage theory. Zhang et al [18] analyzed the effects of shale hydration from macro and micro perspectives through compression tests and CT scan tests. In terms of temperature disturbance: Park et al. [19] conducted freeze-thaw cycling tests on rock specimens and used X-ray computed tomography (CT) and scanning electron microscopy (SEM) to obtain images of microstructural changes within the rock, as well as to measure the changes in the physical properties, and investigated the effects of freeze-thawing of water within the pore spaces, cracks, and seams of the rock on the microstructure and physical properties of the rock. Wang et al. [20] conducted a detailed investigation of the mechanical behavior of rocks after temperature disturbance. In terms of multifactorial perturbations: Ma et al. [21] developed a mathematical model to analyze the stability of the wellbore for the stresses caused by mechanical, hydraulic, and chemical actions, and analyzed the causes of collapse in shale formations. Ekbote and Abousleiman [22] developed a multi-field coupled stress field model for the well wall, but it was not directly applied to evaluate the stability of the well wall. It has also been investigated based on fracture mechanics how factors such as crack angle, circumferential pressure and material properties affect the stress field, displacement field, plastic zone size and crack extension direction [23].

In this regard, scholars have used various methods to carry out disturbance experiments on rock specimens, which can be categorized into physical simulation tests, numerical simulation tests, and derivation of theories. There are many ways of physical simulation tests, such as triaxial compression test [24–27], uniaxial compression test [28,29], power impact test [30–32], $CO_2$ fracturing tests [33,34], etc. The numerical simulation test is a more mainstream test means [35–41]. It is also possible to combine experimental and numerical simulations, and then determine the accuracy of the experimental simulations through theoretical derivation [42–45]. However, post-test detection methods are less innovative, usually acoustic emission [46–49], CT scanning technology [50,51], nuclear magnetic resonance (NMR) techniques [52], and electron microscope scanning techniques [53] are used, The results of the tests and inspections were not analyzed using digital processing.

In summary, many scholars have achieved rich results in the research related to the disturbance of rocks by various factors. However, fewer test results have been digitized. In this paper, the deep shale is studied, and first, their composition and surface microstructure were examined; Second, a stress-controlled triaxial instrument was used to simulate the unloading disturbance effect on the rock; Then, the variation rules of microstructural and mechanical properties of rocks under unloading conditions of different magnitudes were analyzed by SEM and NMR; Finally, damage mechanics and statistical strength theory are combined to model the statistical damage variables of rock unloading disturbances.

## 2 Experimental program

### 2.1 Experimental instruments and processes

According to reference [54]. The main instruments of the test are: MTS electro-hydraulic servo rock testing machine, utilizing a computer control system, applies load to the specimen through a motor-driven hydraulic pump, at the same time measures the deformation of the specimen so as to derive the specimen's mechanical property parameters; JSM-6700F SEM, which is a scanning electron microscope with high resolution, high magnification and excellent image quality, can directly observe the microstructure of the specimen surface; MiniMR-60 Nuclear Magnetic Resonance Instrument, which can carry out radiation-free, non-destructive testing of specimens, etc., and is simple to operate, fast, clear imaging, etc.; XRD-7000 X diffractometer, with X-ray tube emitted X-rays irradiated to the specimen will produce diffraction phenomenon, and then through the radiation detector to accept the diffraction line of the X-ray photons, the formation of diffraction pattern, after the software processing to get the test pattern.

The samples were taken from the gray-black shale of the Longmaxi Formation in the JiaoShiBa area at a plumb depth of 2330-2410m, This formation is a destination reservoir for shale gas aggregation. (1) To make sure the specimen is at the same level, its composition and microstructure are analyzed. (2) Stress control is used to simulate the unloading

perturbation of the well wall surrounding rock, The experimental results were also analyzed by electron microscope scanning and nuclear magnetic resonance. (3) Testing of shale specimens after disturbance with different unloading amplitudes by triaxial compression tests, analyze the deterioration behavior of rock mechanical properties of shale specimens. The flow chart of the test is shown in Fig 1

## 2.2 Experimental study of the components and microstructure of the specimen

Fig 2 displays the $T_2$ energy spectrum distribution, and Fig 3 shows the number of sampling groups for eight groups, numbered N-1~8, to obtain the rock samples of different mineral groups' content. The XRD-7000 type X diffractometer and nuclear magnetic resonance instrument were used to test the sample taken for sampling.

As can be seen in Fig 2, the main mineral components of the shale specimens are clay minerals such as ilmenite/montmorillonite, chlorite, and kaolinite, with contents ranging from 48.8 to 56.4%; quartz, feldspar, and pyrite contents range from 38.6 to 48.2%; carbonatite contents range from 3.6 to 5.7%. The content of the mineral components of each group of specimens is very close to each other, which belongs to the same layer of rock. Fig 3 shows that there are three peaks in the $T_2$ energy spectrum, the first and second peaks are larger in amplitude, and the third peak is almost invisible, and the geometrical pattern of the $T_2$ energy spectra of each group of samples is similar, the amplitude of each peak is close to that, which indicates that the internal pore development of the samples is the same, and they belong to the same stratum of the same rock in the same block.

## 2.3 Experimental design of rock unloading disturbance

In deep geological environments, rocks are typically considered to be under hydrostatic pressure with equal stresses in three orthogonal directions [55]. During drilling operations, the vertical confining pressure on the rock (initially in a hydrostatic state at the excavation face) remains constant, while the horizontal stress progressively decreases due to unloading effects. The tests were conducted using stress control to simulate the unloading disturbance of the well wall perimeter

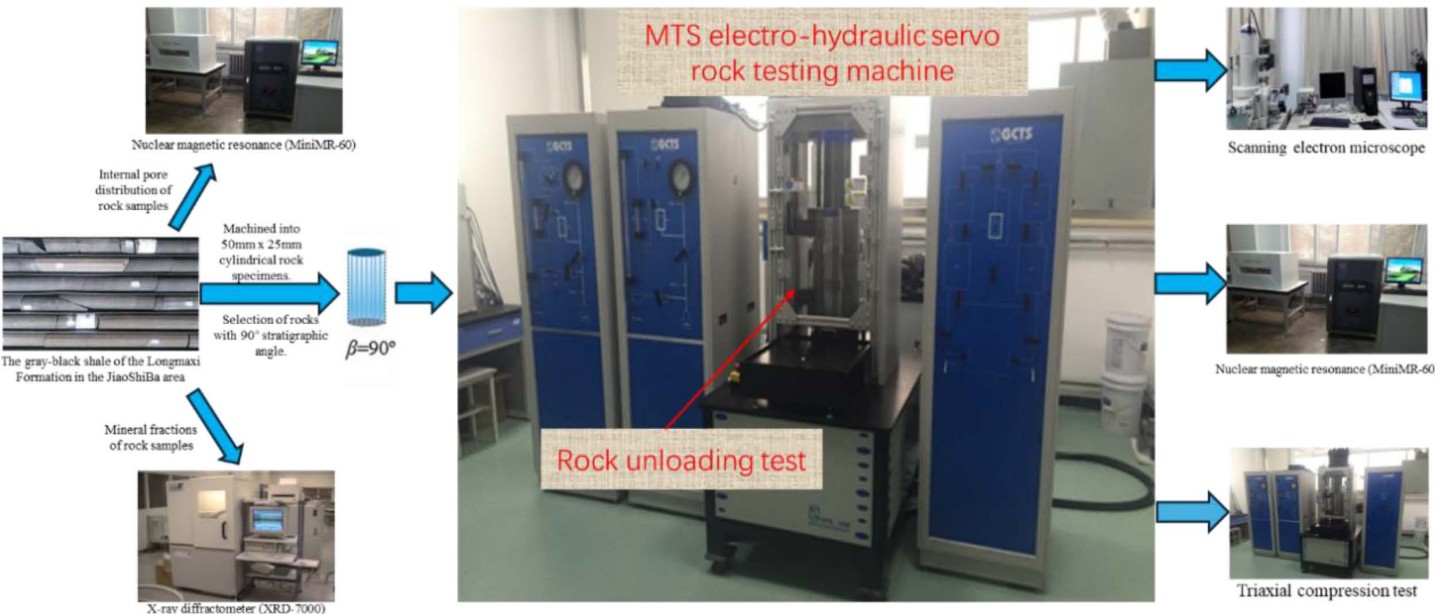

**Fig 1. Experimental procedure flow.**

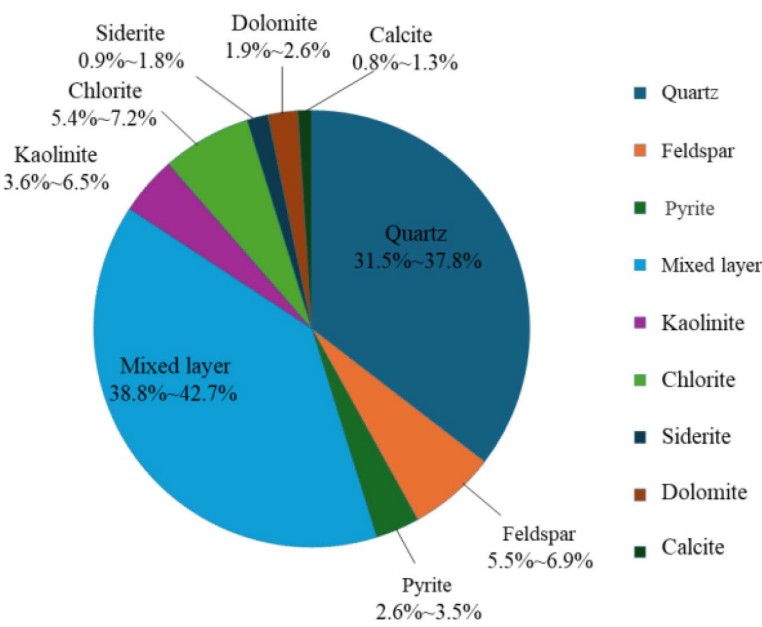

**Fig 2. Mineral fractions of rock samples.**

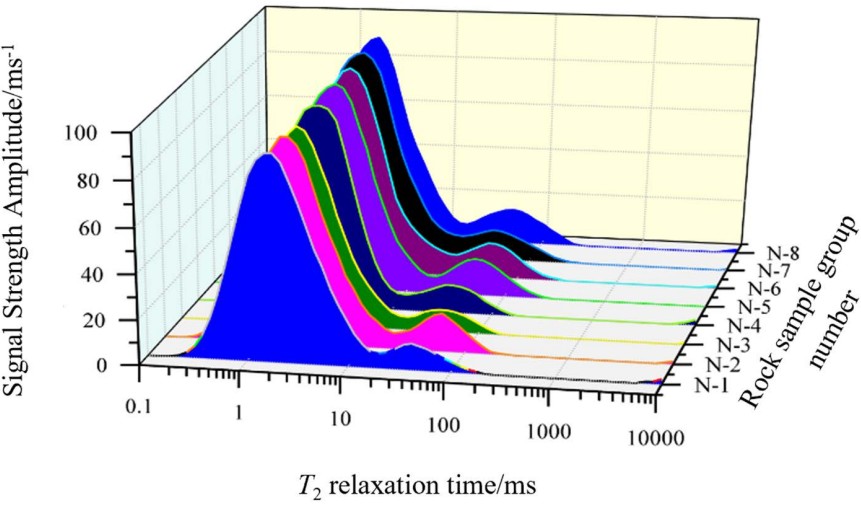

**Fig 3. $T_2$ spectra distribution of samples.**

rock. A shale specimen with a stratigraphic pinch angle of 90º was used for the study. The shale specimens were subjected to unloading simulation tests on a triaxial tester with a 60 MPa enclosure (simulated shale layer burial depth of 2330~2410 m), and then the axial pressure was increased so that the axial stress level approximated to more than 70% of the destructive strength value. In this test, the axial pressure ($\sigma_1$) was designed to be 120 MPa (the peak stress in 90º shale under 60 MPa perimeter rock is 176.82 MPa). The peripheral pressure is then changed while the axial pressure

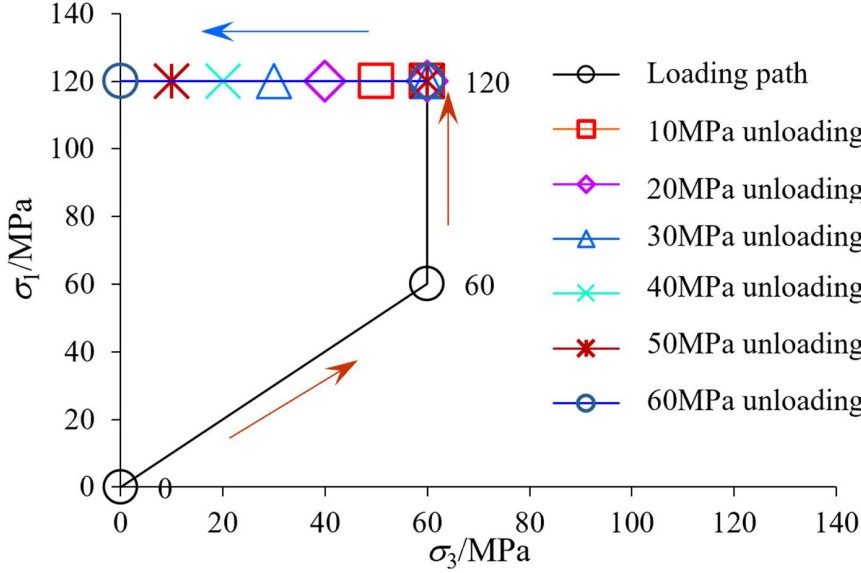

remains constant, with unloading amplitudes of 10 MPa, 20 MPa, 30 MPa, 40 MPa, 50 MPa, and 60 MPa, as illustrated in Fig 4. In the test, the loading and unloading rates were equal to 0.2 MPa/s.

## 3 Experimental studies

### 3.1 Analysis of scanning electron microscope image

Excerpts of SEM images of rock samples subjected to different unloading amplitudes are shown in Fig 5. Among them, the rock samples with an unloading amplitude of 10 MPa were magnified 90 times, the rock samples with unloading amplitude of 20 MPa were magnified 1600 times, the rock samples with unloading amplitude of 30 MPa were magnified 1416 times, the rock samples with unloading amplitude of 40 MPa were magnified 3000 times, the rock samples with unloading amplitude of 50 MPa were magnified 1600 times, and the rock samples with unloading amplitude of 60 MPa were magnified 2400 times.

From Fig 5, the expansion of stress unloading disturbed pores and cracks mainly occurs at the internal large pores or the walls of the solution pores, because the stress concentration occurs at the large pores or the walls of the solution pores during the stress unloading process. Figs 5bc also shows essentially no obvious signs of nascent cracking, again due to the small magnitude of unloading; Fig 5d–f local large solution hole wall or near the region to produce signs of nascent cracks, and with the unloading amplitude increases nascent cracks have increased, there is a tendency to expand, the cause of which is caused by shear extension cracks, As the horizontal principal stresses are unloaded, but the overlying formation pressure remains constant, in this case, the shear force $q = \sigma_1 - \sigma_3$, where $\sigma_1$ is unchanged, $\sigma_3$ reduced, $q$ enlarged, As $\sigma_3$ decreases the amplitude value the greater the increase in $q$, After considering the stress concentration at the pore wall again, shear damage occurs at the pore wall when the shear force q is greater than the shear strength of the local pore wall rock.

Fig 6a–f shows the results of digital image processing of the gray-scale images of SEM in Fig 5a–f using MTALAB software. It was decided to combine the pore structure with the numerical data to obtain more accurate information about the pore structure of shale specimens. As can be seen from Fig 8a–f, the image quality is improved and the matrix and

**Fig 4. Schematic diagram of test stress unloading path.**

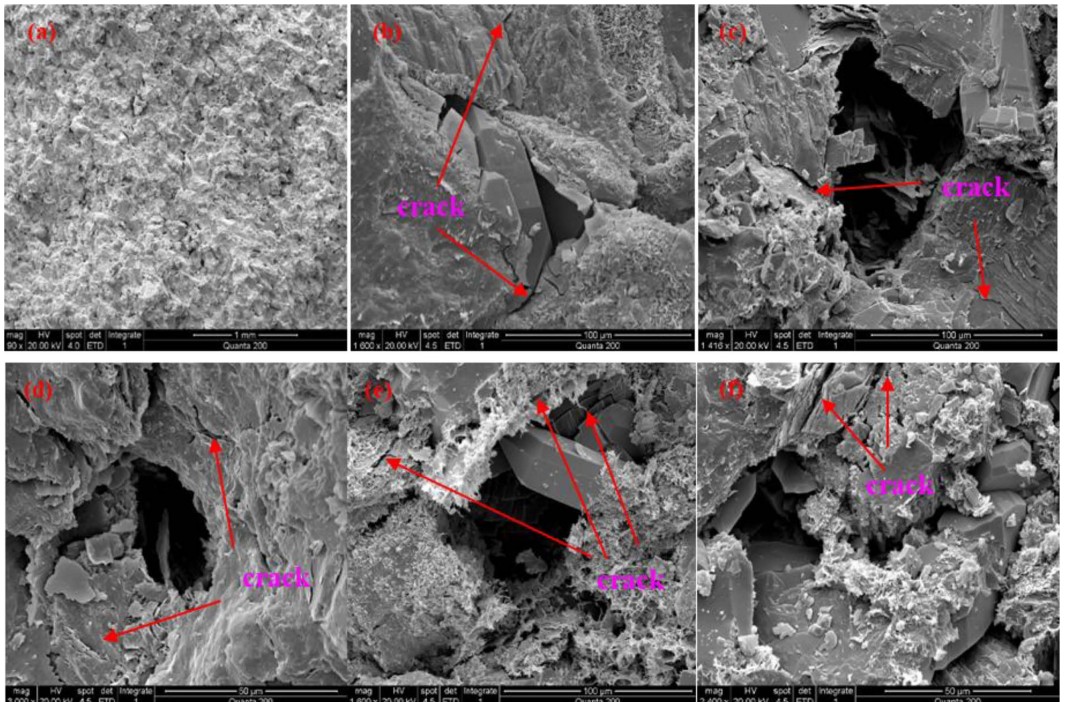

**Fig 5. Images of SEM after unloading disturbance.** Note: (a) The unloading amplitude is 10MPa; (b) The unloading amplitude is 20MPa; (c) The unloading amplitude is 30MPa; (d) The unloading amplitude is 40MPa; (e) The unloading amplitude is 50MPa; (f) The unloading amplitude is 60MPa.

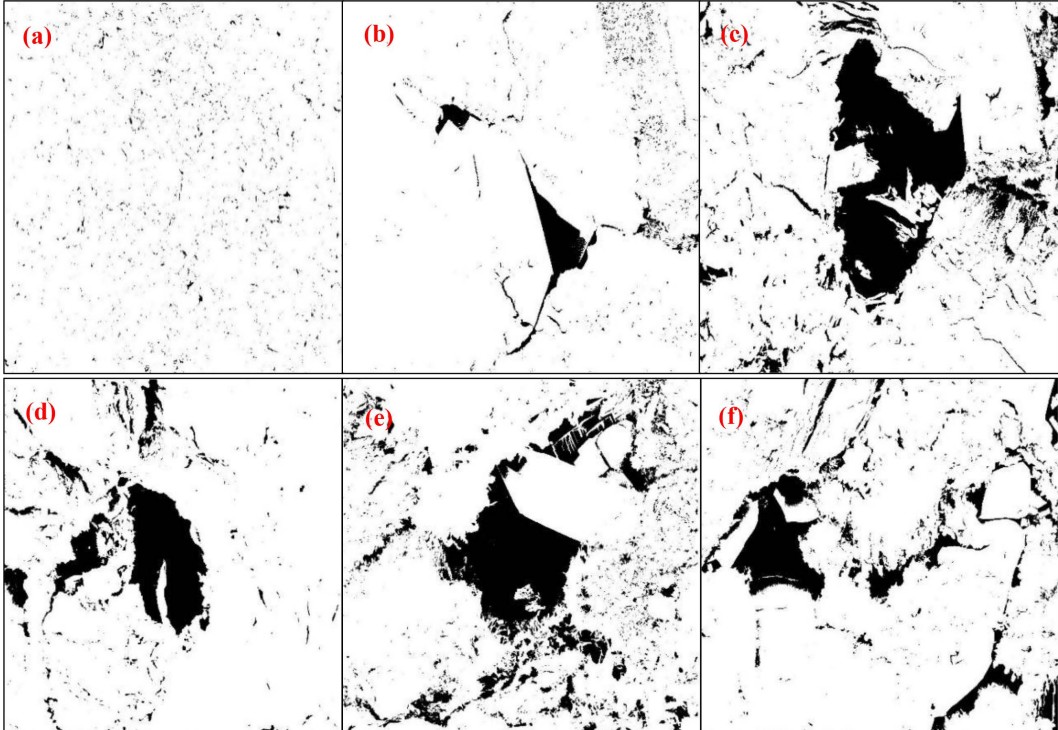

**Fig 6. Images of SEM after binary processing.**

pore boundaries in the shale specimen are more clearly distinguished by sharpening, binary segmentation, and noise reduction.

Therefore, the shale porosity and pore ratio may be described as follows: The black region in Fig 6 depicts the shale surface defects, such as pores, holes, and fractures, whereas the white high-brightness area represents the shale skeleton part.

$$n = \frac{S_p}{S_m + S_p} \times 100\%$$

(1)

$$e = \frac{S_p}{S_m}$$

(2)

Where: $S_p$ is the area of black area; $S_m$ is the area of white high brightness area.

The principle of image digitization states that sampling operations on the coordinates and amplitude are necessary to convert a continuous image into digital form; quantization is the process of digitizing the amplitude values, and sampling is the process of digitizing the coordinate values. Fig 7a illustrates how the sensor that created the image typically acquires the image sampling and projects it onto the sensor array. Fig 7b illustrates how image quantization entails discretizing the array of sampled interior parts and allocating a gray value based on the white brightness of each discrete grid, thereby quantizing the continuous gray values at various locations of the image into discrete quantities.

Based on the principle of image digitization, the Representation of a digitized image as a two-dimensional function $f(x, y)$, x, y are the image width and height side coordinates, the magnitude of the function represents the brightness at any ($x$, $y$) and the image matrix representation is shown in Eq. (3).

$$f(x, y) = \begin{bmatrix} f(0, 0) & f(0, 1) & \cdots & f(0, N-1) \\ f(1, 0) & f(1, 1) & \cdots & f(1, N-1) \\ \cdots & \cdots & \cdots & \cdots \\ f(M-1, 0) & f(M-1, 1) & \cdots & f(M-1, N-1) \end{bmatrix}$$

(3)

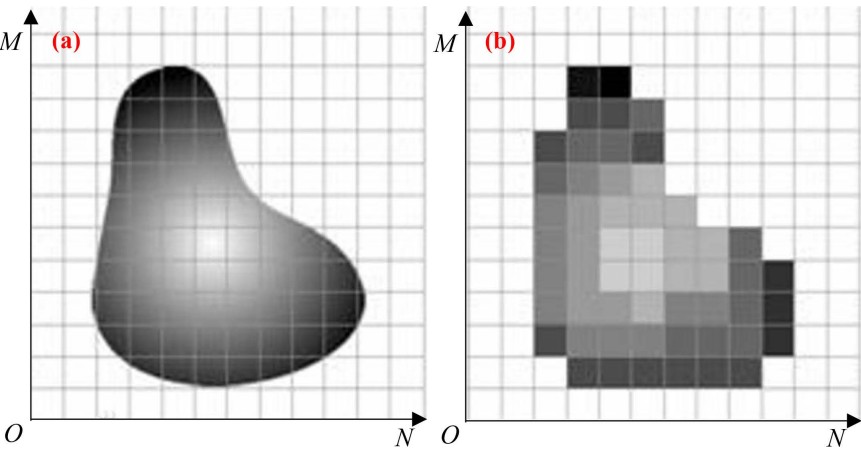

**Fig 7. Image digitization process.** Note: (a) Continuous image that has been projected onto the sensor array; (b) Results of image sampling and quantization.

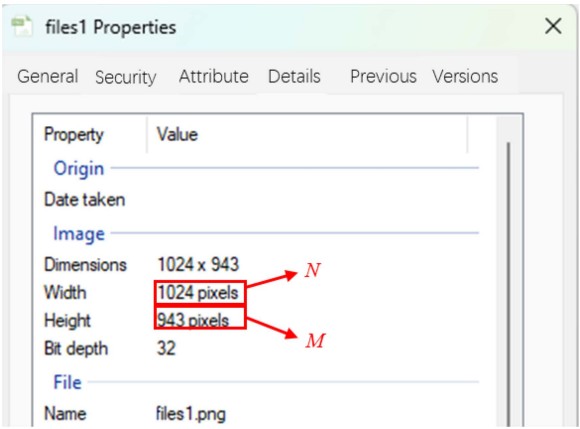

**Fig 8. Image details.**

Where: $N$ is the number of pixels on the wide side of the image, $M$ is the number of pixels on the high side of the image, the size of $N$ and $M$ determines the resolution of the image, the larger the value, the clearer the image, the size of the value of $N$ and $M$ can be given by the detailed information of the image as shown in Fig 8, i.e., $N=1024$, $M=943$.

According to reference [56]. Logical arrays that only accept values between 0 and 1 are known as binarized pictures. By digitizing the size of the gray value of each grid position in the grayscale image, the specific acquisition process consists of defining a logical array of 0 and 1. In other words, if the gray value is less than or equal to a given number (in this paper, it is taken as 50), take the value of 0, and if the gray value is greater than a certain number (in this paper, it is taken as 50), take the value of 1. The following value is thus obtained by the amplitude $f(x_i, y_j)$ at any point in Eq. (3).

$$f(x_i, y_j) = \begin{cases} 0 & g \leq 50 \\ 1 & g > 50 \end{cases}$$

(4)

Where: g denotes the gray value, i.e., the magnitude size of $f(x_i, y_j)$ amplitude, $i$ takes values from 0 to $M$-1, and $j$ takes values from 0 to $N$-1. When $i=0$ and $j=0$, it denotes the origin.

Then the matrix of Eq. (3) will change to a matrix with each element taking the value 0 or 1. A value of 0 for $f(x_i, y_j)$ indicates that the region is black, then $S_p$ can be superimposed with each region that takes the value 0; A value of 1 indicates that the area is white, and $S_m$ can be overlaid with each area that takes the value 1. Since each $f(x_i, y_j)$ corresponds to an area of $S_{ij}$, finding the area is converted to counting the number of elements that take the value 0 or 1.

$$N_1 = \sum_{i=0}^{N-1} \sum_{j=0}^{M-1} f(x_i, y_j), \; g > 50$$

(5)

$$N_0 = N \times M - N_1$$

(6)

Then $S_p$ and $S_m$ can be expressed as

$$S_P = N_0 \times S_{ij}$$

(7)

$$S_m = N_1 \times S_{ij}$$

(8)

Then Eqs (1) and (2) change to

$$n = \frac{N_0}{N_1 + N_0} \times 100\%$$

(9)

$$e = \frac{N_0}{N_1}$$

(10)

The calculation methods of Eqs. (9) and (10) can be used to derive the magnitude of porosity and pore ratio for each figure in Fig 6, as shown in Table 1.

The porosity of the shale, as determined from Figs 6a–f, differs significantly, as Table 1 shows, suggesting that the distribution and growth of pores and fractures on the shale's surface are irregular. The porosity of the shale overall specimen surface pores, as shown in Fig 6a, is small and more representative of the entire pore development; however, there are flaws because of the low image magnification, which causes the pores to be counted incompletely, which leads to a low porosity calculation; Additionally, the specimen's localized regions with noticeable pore, fracture, and solution hole formation are shown in Figs 6b through 6f. Their surface porosity is 1.81~7.3 times greater than the specimen's overall porosity. These pores, fractures, and lysimetric pores are mostly seen in the clay mineral distribution region, which more recently has shown that there is a clustering phenomenon in the shale mineral distribution as well.

Quantitative microstructure analysis of the pictures based on image processing using MTALAB software was conducted using Image-Pro Plus 6.0 (IPP 6.0). To determine the units of measurement associated with an image and to create a spatial scale of measurement associated with the magnified image, the method's basic idea is to apply a calibrated spatial scale to the image at a certain magnification. The IPP 6.0 software converts all space measurements from pixels to microns. Specific processing steps: (1) Setting the pixel ratio, since the image is calculated in pixels and the spatial dimensions are defined in terms of length; therefore, the length is defined according to the scale of the SEM image, and the conversion of pixels to length units is performed. (2) Selection and calculation of microscopic pore structure parameters, characterizing the microscopic pore structure parameters are pore number, porosity, area, maximum pore diameter, average pore diameter, minimum pore diameter,etc. (3) Based on the statistical analysis of the pore structure parameters of the selected areas of the images, the microscopic pore structure parameters are obtained, and the microscopic pore structure of the shale can be quantitatively analyzed.

The fine pore characterization parameters, including the number of pore cracks, the maximum area, the minimum area, the maximum pore diameter, the minimum pore diameter, and other shale microporosity size dimensions and number of shales microporosities, will also be computed and counted using the IPP 6.0 image analysis software in the six binarized images in Fig 6a–f. Figs 9 and 10 schematically depict the particular statistical procedure and characterization parameters, whereas Table 2 displays the statistical outcomes determined by this program.

According to Table 2's statistical results, the minimum area in the microfine view pore parameters of the rock varies in the statistics because of the various SEM magnifications. For example, Fig 6a shows that at 90 times magnification,

**Table 1.  Porosity and void ratio of each binary image.**

| Binarization chart | Fig 6 | | | | | |
|---|---|---|---|---|---|---|
| Pore index | (a) | (b) | (c) | (d) | (e) | (f) |
| Magnification/times | 90 | 1600 | 1416 | 3000 | 1600 | 2400 |
| Porosity/% | 1.82 | 3.30 | 13.28 | 9.87 | 11.02 | 11.85 |
| Porosity ratio | 0.0185 | 0.0341 | 0.1531 | 0.1095 | 0.1238 | 0.1344 |

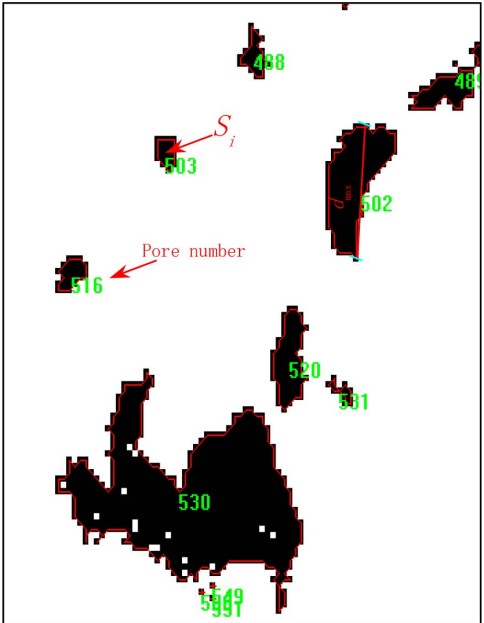

**Fig 9. Statistics of pore structure parameters in selected areas.**

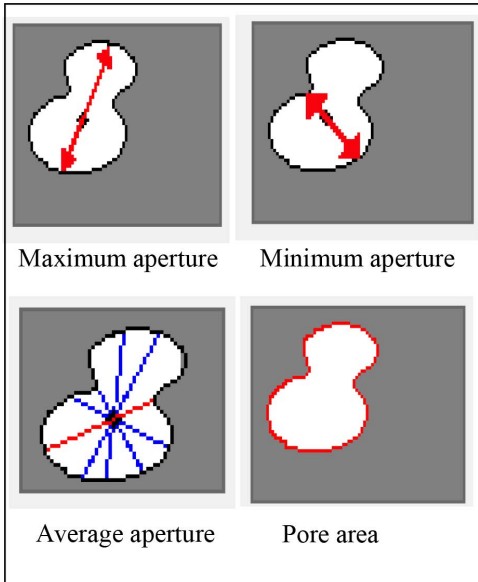

Maximum aperture Minimum aperture

Average aperture Pore area

**Fig 10. Aperture parameters.**

the minimum area of the statistics is 115 μm², while Figs 6b–f shows that at 1416~3000 times magnification, the minimum area of the statistics is 10~13 μm². Similarly, the existence of significant lysimeters or voids in the scanned region causes variations in the maximum area measured. Controlling the pore size (pore area) in the same region for statistical analysis and comparison is required to investigate the evolution of the rock's interior pores; that is, the range of pore area

**Table 2. Microcosmic pores parameters calculation results.**

| Parameters \\ Imagery | Fig 6 | | | | | |
|---|---|---|---|---|---|---|
| | (a) | (b) | (c) | (d) | (e) | (f) |
| Magnification | 90 | 1600 | 1416 | 3000 | 1600 | 2400 |
| Number of pores | 243 | 243 | 434 | 267 | 649 | 325 |
| Minimum area/μm² | 115 | 13 | 10 | 10 | 10 | 10 |
| Maximum area/μm² | 14452 | 12354 | 10195 | 12786 | 11597 | 10095 |
| Total area/μm² | 480192 | 87654 | 66275 | 70356 | 98951 | 55917 |
| Maximum aperture/μm | 167.3 | 96.2 | 167.6 | 147.3 | 156.9 | 118.9 |
| Minimum aperture/μm | 10.5 | 5.1 | 3.6 | 3.9 | 3.8 | 3.7 |

comparison is 13–10095 μm². The range is the aperture that contains all the statistics in the image. Since the results of the rock pore parameters counted in Fig 6a are not in the control range, they will not be analyzed for comparison.

The relationship between the maximum pore size and the minimum pore size of each pore of the microfine view pore according to the IPP 6.0 software statistics. According to IPP 6.0, software statistics of the relationship between the maximum pore size and minimum pore size of each pore of the microfine view pore, $d_{max}/d_{min}$ is used to define the holes (voids) and cracks, if $d_{max}/d_{min} \leq 3$ is the holes (voids), and $d_{max}/d_{min} > 3$ is the cracks. Specifically to count the number of pores with $d_{max}/d_{min}$ ratio in the range of [1, 2], (2, 3], and (3, +∞) and the percentage of the total number of pores, which statistics can be used to analyze the existence of shale rock pores in the form of holes, or cracks, as shown in Table 3 and Fig 11.

**Table 3. Percentage of different $d_{max}/d_{min}$.**

| $d_{max}/d_{min}$ \\ Imagery | [1, 2] | | | | (2, 3] | | | | (3, +∞) | | | |
|---|---|---|---|---|---|---|---|---|---|---|---|---|
| | Number/pc (%) | | Area/μm² (%) | | Number/pc (%) | | Area/μm² (%) | | Number/pc (%) | | Area/μm² (%) | |
| Fig 6a | 106 | 43.6 | 202785 | 42.2 | 86 | 35.4 | 164207 | 34.2 | 51 | 21 | 113200 | 23.6 |
| Fig 6b | 82 | 36.5 | 10731 | 27.9 | 77 | 34.3 | 10269 | 26.7 | 66 | 29.2 | 17501 | 45.5 |
| Fig 6c | 141 | 32.6 | 8525 | 19.3 | 141 | 32.6 | 8128 | 18.4 | 150 | 34.7 | 27521 | 62.3 |
| Fig 6d | 83 | 31.4 | 5198 | 13.0 | 85 | 32.2 | 6198 | 15.5 | 96 | 36.4 | 28592 | 71.5 |
| Fig 6e | 225 | 34.8 | 10468 | 13.6 | 204 | 31.5 | 15318 | 19.9 | 318 | 33.6 | 51189 | 66.5 |
| Fig 6f | 97 | 29.8 | 2628 | 4.7 | 103 | 31.7 | 15433 | 27.6 | 125 | 38.5 | 37799 | 67.6 |

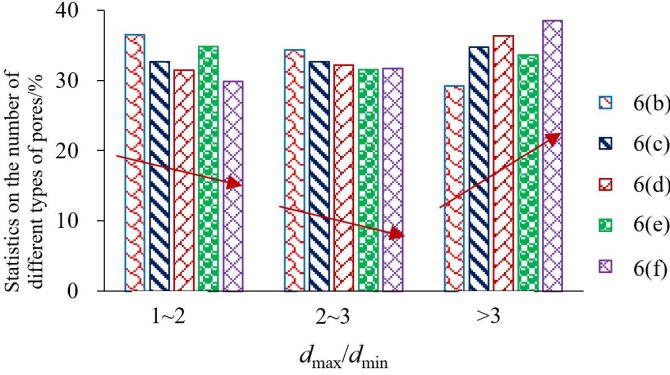

**Fig 11. Number percentage of pores with different $d_{max}/d_{min}$ ratios.**

As can be seen from <u>Fig 11</u>, with the increase of unloading amplitude, regardless of the magnification of the image, there is an overall decrease in the number of pores with $d_{max}/d_{min}$ ratios between [1, 2], indicating that there is a tendency to expand the development of such pores with the increase of unloading amplitude, i.e., some of the pores have a shift from $d_{max}/d_{min}$ ratios in [1, 2] to (2, 3]. Similarly, there is a decrease in the number of pores with $d_{max}/d_{min}$ ratio between (2, 3], which also indicates that there is a shift from $d_{max}/d_{min}$ ratio (2, 3] to (3, +∞) with increasing unloading amplitude in some of the pores, and indicates that there is a tendency for the pores to expand and develop. Finally, the number of pores with $d_{max}/d_{min}$ ratios between (3, +∞) increases with the unloading amplitude, indicating that the number of such pores increases significantly, but has not yet reached the stage of expansion and development of convergence and penetration. The total overall analysis shows that the unloading disturbance is just the beginning of crack initiation and small expansion of the local microcracks.

From the analysis of the evolution of area expansion, it can be seen from <u>Fig 12</u> that there is a sudden increase in the area of this type of pore firstly and also analyzing $d_{max}/d_{min}$ between (3, +∞), indicating that the unloading disturbance pore expansion is also mainly focused on the development of dominant pores; Further analysis of such pores with $d_{max}/d_{min}$ between (2, 3] reveals that they first decrease and then start to grow, indicating that with the increase in unloading magnitude, such pores with $d_{max}/d_{min}$ between [1, 2] start to expand and develop progressively. Such pores with $d_{max}/d_{min}$ between [1, 2] have a mega decrease condition, indicating that almost no new pores are sprouting and there is an expansion of this type of pores. During stress unloading in surrounding rock, macroscopic fractures demonstrate the highest susceptibility to crack initiation, propagation, interconnection, and localized damage evolution. This mechanism facilitates the full development of damage along potential slip surfaces, with shale specimens exhibiting pronounced brittle failure characteristics. Furthermore, immediately after borehole excavation (before mudcake formation), the wellbore lacks effective wall support. Only limited stabilization is provided by near-wellbore seepage resistance – a mechanism contingent on well-developed formation permeability, yet typically insufficient due to minimal seepage constraints. Consequently, stress redistribution occurs radially from the wellbore into the surrounding rock during this pre-mud cake phase, inducing stress relief perturbations in the near-wellbore zone.

### 3.2 Analysis of NMR test results

The distribution patterns of $T_2$ energy spectra of rock samples at different unloading amplitudes were measured by NMR tests, as shown in <u>Fig 13</u>. As can be seen from <u>Fig 13</u>, (1) the initial rock samples have a high number of micro and small pores and a low number of large and medium pores. With different degrees of unloading $T_2$ energy spectrum peaks have a tendency to increase, but the magnitude of the increase is different, the first peak did not increase much, indicating that

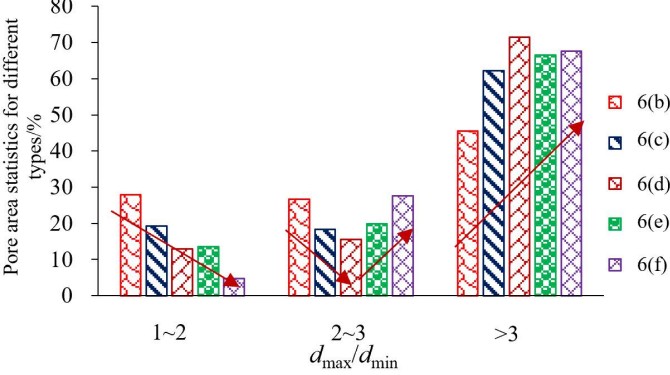

**Fig 12. Area percentage of pores with different $d_{max}/d_{min}$ ratios.**

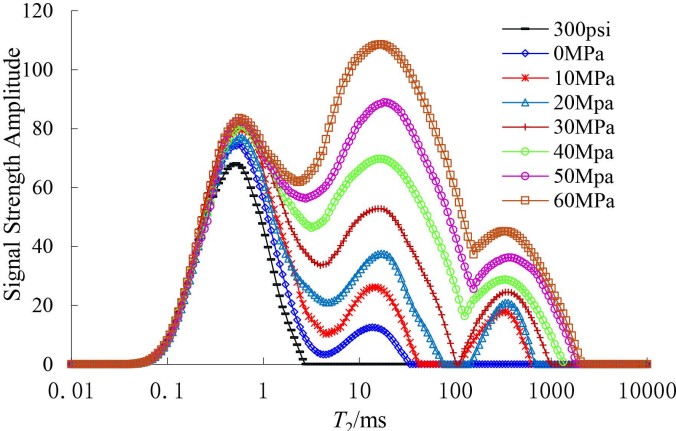

**Fig 13. Shale $T_2$ energy spectrum with different unloading amplitudes.**

**Table 4. Statistical parameters of $T_2$ energy spectrum under different unloading amplitudes.**

| Unloading amplitude/MPa | | 0 | 10 | 20 | 30 | 40 | 50 | 60 |
|---|---|---|---|---|---|---|---|---|
| Spectral area | Microporous | 1725.6 | 1863.3 | 1750.9 | 1935.9 | 1842.8 | 1921.3 | 1930.8 |
| | Mesopore | 692.1 | 1248 | 1742.3 | 2717.3 | 3745.3 | 4584 | 5613.7 |
| | Macroporous | 0 | 275.4 | 292.2 | 496.8 | 784.8 | 1143.1 | 1534 |
| Total spectral area | | 2417.7 | 3386.6 | 3785.4 | 5149.9 | 6373 | 7648.4 | 9078.5 |
| Growth rate of spectral area/% | | / | 40.08 | 56.57 | 113.01 | 163.6 | 216.35 | 275.5 |
| The proportion of peaks in each type of pore spectrum/% | Microporous | 71.37 | 55.02 | 46.25 | 37.59 | 28.92 | 25.12 | 21.27 |
| | Mesopore | 28.63 | 36.85 | 46.03 | 52.76 | 58.77 | 59.93 | 61.83 |
| | Macroporous | 0 | 8.13 | 7.72 | 9.65 | 12.32 | 14.95 | 16.9 |

there is not much participation in the development and expansion between the tiny pores; (2) Both the second and third peaks showed a large increase, especially the second peak increased 8~9 times the original peak height, and the third peak, initially absent, increased with the unloading amplitude, and finally increased to a signal strength of 43.59, It shows that the unloading process mainly leads to the development and expansion of medium and large pores, and the pores are interconnected with each other, and some of the micro and small pores are transformed into medium and large pores. The significant change in the number of medium and large pores is also an indication that the large and medium pores belong to the extended dominant pores, and the tiny pores are still associated with a more stable structure under loading.

Table 4 lists the $T_2$ energy spectrum areas that correspond to each type of pore. Fig 14 illustrates the growth pattern of the $T_2$ energy spectrum region with increasing unloading amplitude. As well as the proportion and evolutionary pattern of each kind of pore space in the rock with unloading amplitude, as seen in Figs 15 and [16].

Fig 14 illustrates how the NMR energy spectral area increases as the unloading amplitude increases. This growth is almost linear, and the rate of increase of the energy spectral area is as follows: 10 MPa with a growth rate of 40.08%, 20 MPa with a growth rate of 56.57%, 30 MPa with a growth rate of 113%, 40 MPa with a growth rate of 163.6%, 50 MPa with a growth rate of 216.35%, and 60 MPa with a growth rate of 275.5% are the unloading amplitudes.

As can be seen from Fig 15, with the increase of unloading amplitude, the proportion of mesopore and macroporous energy spectral area are both increased, in which the rate of increase of mesopore is faster, and the proportion of microporous energy spectral area is relatively lower, and the overall change is relatively smooth and close to a straight line. The

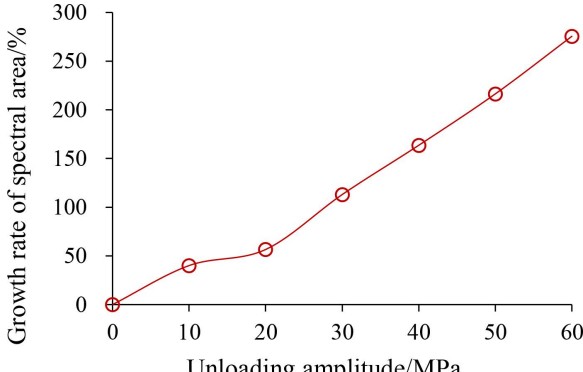

**Fig 14.** $T_2$ **energy spectrum area growth rate after pores after different unloading amplitudes.**

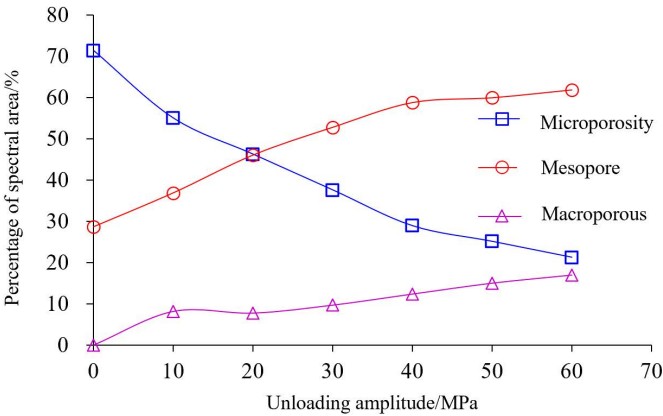

**Fig 15. Proportion of various unloading amplitudes.**

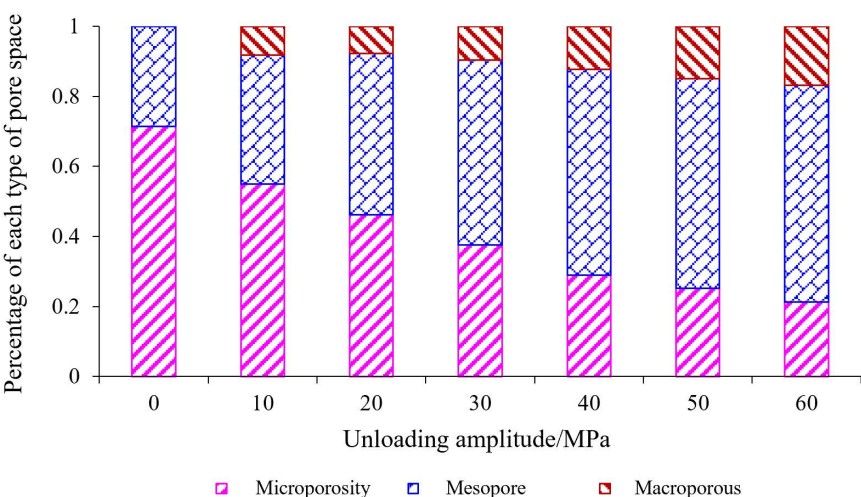

**Fig 16. Percent stacking chart of micropore, mesopore, and macropore spectrum area.**

increase in the area of both mesoporous and macroporous energy spectra indicates an increase in the number of both mesoporous and macroporous pores, and a significant increase in the number of mesoporous and porosities, suggesting that the unloading disturbance is mainly focused on the expansion and development of tiny and mesoporous pores on the walls of the large voids or at shallow depths.

As can be seen from Fig 16, with the increase of unloading amplitude, the proportion of different types of pores showed different degrees of changes, and the proportion of micropores was 71.37% when there was no unloading, and finally decreased to 21.27%; Medium porosity and large porosity have increased, medium porosity increased most significantly from 28.63% to 61.83%, the proportion of large porosity also has a slow increase, it can be seen that the evolution of rock porosity, damage process, pore size conversion is relatively smooth.

Considering the impact of diffusive relaxation, the transverse relaxation time $T_2$ is frequently employed as a gauge of porosity in rocky materials. The water-wet rock is mostly impacted by surface fluid relaxation when the external magnetic field is weak, or the wave interval $T_E$ is brief. The size of this relaxation may be calculated using the pore's specific surface area, or $T_2$, which can be written as follows:

$$\frac{1}{T_2} = \rho_2 \frac{S}{V_p}$$

(11)

Since $S/V_p$ is related to the pore throat radius r as in Eq. (12)

$$\frac{S}{V_p} = \frac{F_s}{r}$$

(12)

Then Eq. (11) can be simplified as:

$$\frac{1}{T_2} = F_s \frac{\rho_2}{r}$$

(13)

where $F_s$ is the geometry factor; $r$ is the pore radius.

According to the correspondence between the pore radius $r$ and the transverse relaxation time $T_2$ of the rock sample in Eq. (13), for columnar pores, $F_s$ is taken as 2; The $T_2$ surface relaxation strength is generally taken as $0.5 \times 10^{-8}$ ms$^{-1}$; then the equation for calculating the pore radius $r$ can be obtained.

$$r = 10^{-8} T_2$$

(14)

From Eq. (14), there is a consistency between the distribution curve of transverse relaxation time $T_2$ and the distribution curve of rock sample pore size: The larger the $T_2$, the larger the pore size; the higher the $T_2$ amplitude, the higher the number of pores corresponding to the pore size. The shale lateral relaxation time $T_2$ distribution curve can be transformed into a pore radius distribution curve, which will classify the shale internal pores into three categories according to the NMR energy spectrum: Pore sizes between $10^{-4}$ μm ≤ $r$ < $10^{-2}$ μm (0.01 ms ≤ $T_2$ < 1 ms) are considered microporous; pore sizes between $10^{-2}$ nm ≤ $r$ < 1 μm (1 ms ≤ $T_2$ < 100 ms) are medium pores; pore sizes between 1 μm ≤ $r$ < 100 μm (100 ms ≤ $T_2$ < 10000 ms) are macropores.

The spectral area of the $T_2$ energy spectrum plot visualizes the following relationship: (1) The size of the spectral area is proportional to the amount of fluid contained inside the rock after saturation, i.e., the size of the spectral area is related to the porosity of the rock; (2) The size of the spectral areas of the different peaks indicates that the energy spectral areas of the peaks corresponding to the microporosity, mesoporosity, and macroporosity of the water-saturated rock (i.e., the

size of the spectral areas of the first, second, and third peaks) are directly proportional to the total number of microporosities, mesoporosities, and macroporosities, respectively.

Based on the above analysis of the relationship between $T_2$ energy spectrum area and porosity, water is used as the flowable fluid in this paper. The NMR test was performed to test the $T_2$ energy spectra of the rock samples in the saturated state, and then the saturated rock samples were placed in a 300 psi centrifuge to expel the flowable fluid, and then the $T_2$ energy spectra of the rock samples after centrifugation were tested and compared with the $T_2$ energy spectra of the rock samples in the saturated state, as shown in Fig 17.

As can be seen from Fig 17, the $T_2$ energy spectrum area corresponding to the area of the shaded portion is the pore volume occupied by the flowable water belonging to the centrifuge dumping, and a relationship can be established between the total pore volume ($V_p$), the pore volume occupied by the centrifuge dumping water ($\Delta V_w$), the area of the $T_2$ energy spectrum in the saturated state ($S_B$), and the area of the shaded portion ($S_Y$) as shown in Eq. (15).

$$\frac{\Delta V_w}{V_P} = k\frac{S_Y}{S_B}$$

(15)

where: k is the scale factor.

Assuming k is a constant, determine the pore volume and porosity containing the value of k. In solving for the pore increment multiplier, the relationship between the $T_2$ spectral area and porosity can be established by approximating it with the scaling factor k.

$T_2$ energy spectrum area finding method: In the $T_2$ energy spectrum test, the energy spectrum is divided into 200 equal parts on the $T_2$ axis (which has been determined at the time of the test), the length of each equal part is noted as $\Delta T_2$ (constant), and the value of the signal intensity of each equal part of the $T_2$ energy spectrum is noted as:

$$S(T_2) = \sum_{i=1}^{200} \Delta T_2 \cdot \rho_i(T_2) = \Delta T_2 \sum_{i=1}^{200} \rho_i(T_2)$$

(16)

The mass ($\Delta m_w$) and volume ($\Delta V_R$) of flowable water can be calculated by weighing the rock sample after centrifugation at 300 psi.

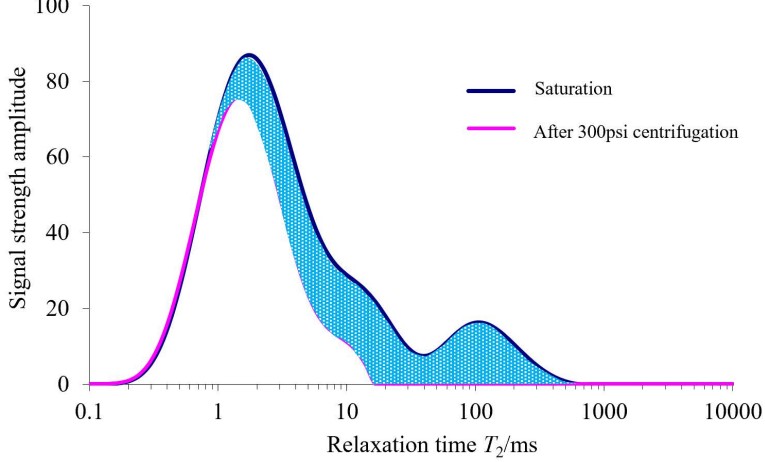

**Fig 17. $T_2$ spectrum distribution in the saturated state and after centrifugation.**

$$\Delta m_{\mathrm{w}} = m_{\mathrm{S}} - m_{\mathrm{R}} \tag{17}$$

$$\Delta V_{\mathrm{w}} = \frac{\Delta m_{\mathrm{w}}}{\rho_{\mathrm{w}}} \tag{18}$$

Where: $m_{\mathrm{s}}$ is the mass of the saturated specimen, g; $m_{\mathrm{R}}$ is the mass of the specimen after centrifugation, g; and $\Delta V_{\mathrm{w}}$ is the volume of dumped water, cm³. Area of $T_2$ energy spectrum ($S_{\mathrm{B}}$) and area of the shaded portion ($S_{\mathrm{Y}}$) in the saturated state

$$S_{\mathrm{B}} = \Delta T_2 \sum_{i=1}^{200} \rho_{\mathrm{B},i}(T_2) \tag{19}$$

$$S_{\mathrm{Y}} = \Delta T_2 \sum_{i=1}^{200} [\rho_{\mathrm{B},i}(T_2) - \rho_{\mathrm{L},i}(T_2)] \tag{20}$$

Where: $\rho_{\mathrm{B},i}(T_2)$ is the signal intensity of the $T_2$ energy spectrum of the water-saturated rock sample. $\rho_{\mathrm{L},i}(T_2)$ is the $T_2$ energy spectrum signal intensity of the rock sample after initial centrifugation.

Substituting Eqs. (18) to (20) into Eq. (15) yields the pore volume ($V_{\mathrm{p}}$)

$$V_{\mathrm{P}} = \frac{\Delta m_{\mathrm{w}}}{k\rho_{\mathrm{w}} \sum_{i=1}^{200} [\rho_{\mathrm{B},i}(T_2) - \rho_{\mathrm{L},i}(T_2)]} \sum_{i=1}^{200} \rho_{\mathrm{B},i}(T_2) = \frac{\Delta m_{\mathrm{w}}}{k\rho_{\mathrm{w}}} \sum_{i=1}^{200} \left[1 + \frac{\rho_{\mathrm{L},i}(T_2)}{\rho_{\mathrm{B},i}(T_2) - \rho_{\mathrm{L},i}(T_2)}\right] \tag{21}$$

The unloading disturbances in the project cause the rock strain state to change, i.e., there is micro-defect evolution within the rock such as micro-defect sprouting, expansion, and convergence. When NMR testing is used, the stress is released and the rock specimen becomes larger, and it is considered that the volume increment originates from the contribution of the pore volume and the bedrock volume remains unchanged. Then the pore ratio ($e$) can be calculated using the $T_2$ energy spectrum area can be shown by Eq. (22).

$$e = \frac{V_{\mathrm{P}}}{V_{\mathrm{S}}} = \frac{\Delta m_{\mathrm{w}}}{k V_{\mathrm{S}} \rho_{\mathrm{w}}} \sum_{i=1}^{200} \left[1 + \frac{\rho_{\mathrm{L},i}(T_2)}{\rho_{\mathrm{B},i}(T_2) - \rho_{\mathrm{L},i}(T_2)}\right] \tag{22}$$

Where: $V_{\mathrm{S}}$ is the volume of bedrock, cm³.

Similarly, the pore ratio of the rock sample under the initial condition can be obtained as

$$e_0 = \frac{\Delta m_{\mathrm{w}0}}{k V_{\mathrm{S}} \rho_{\mathrm{w}}} \sum_{i=1}^{200} \left[1 + \frac{\rho_{\mathrm{L}0,i}(T_2)}{\rho_{\mathrm{B}0,i}(T_2) - \rho_{\mathrm{L}0,i}(T_2)}\right] \tag{23}$$

Where: $\rho_{\mathrm{B}0,i}(T_2)$ is the initial water-saturated rock sample $T_2$ energy spectrum signal intensity; $\rho_{\mathrm{L}0,i}(T_2)$ is the initial centrifuged rock sample $T_2$ energy spectrum signal intensity; $e_0$ is the initial pore ratio, which can be determined by conventional tests.

Since there is no change in $V_{\mathrm{S}}$ when the strain state changes, Eqs. (22) and (23) can be obtained by collapsing Eq.

$$\frac{\Delta m_{\mathrm{w}}}{k e \rho_{\mathrm{w}}} \sum_{i=1}^{200} \left[1 + \frac{\rho_{\mathrm{L},i}(T_2)}{\rho_{\mathrm{B},i}(T_2) - \rho_{\mathrm{L},i}(T_2)}\right] = \frac{\Delta m_{\mathrm{w}0}}{k e_0 \rho_{\mathrm{w}}} \sum_{i=1}^{200} \left[1 + \frac{\rho_{\mathrm{L}0,i}(T_2)}{\rho_{\mathrm{B}0,i}(T_2) - \rho_{\mathrm{L}0,i}(T_2)}\right] \tag{24}$$

 

After organizing Eq. (24), we can get

$$e = \frac{e_0 \Delta m_\text{w}}{\Delta m_\text{w0}} \sum_{i=1}^{200} \left[ \frac{\rho_{\text{B},i}(T_2)[\rho_{\text{B}0,i}(T_2) - \rho_{\text{L}0,i}(T_2)]}{\rho_{\text{L}0,i}(T_2)[\rho_{\text{B},i}(T_2) - \rho_{\text{L},i}(T_2)]} \right]$$

(25)

Eq. (25) can be used to test the size and law of the change of pore ratio of the rock at any time using nuclear magnetic resonance technology, which is used to calculate and explore the rule of change of the pore ratio of the shale rock in the process of drilling after various types of construction multifactorial perturbation effects.

When $\Delta m_\text{R} - \Delta m_\text{R0} \le 0.1\,\text{g}$, it can be approximated as $\Delta m_\text{R} = \Delta m_\text{R0}$, which is $S_\text{R} = S_\text{R0}$.

The same reference to Eq. (20) yields the area of the shaded portion for the initial condition

$$S_\text{Y0} = \sum_{i=1}^{200} [\rho_{\text{B},i}(T_2) - \rho_{\text{L}0,i}(T_2)]$$

(26)

Comparison of Eqs (20) and (26) shows that

$$\rho_{\text{L},i}(T_2) = \rho_{\text{L}0,i}(T_2)$$

(27)

Substituting Eqs. (27) into (25) yields

$$e = \frac{e_0 \Delta m_\text{w}}{\Delta m_\text{w0}} \sum_{i=1}^{200} \left[ \frac{\rho_{\text{B},i}(T_2)[\rho_{\text{B}0,i}(T_2) - \rho_{\text{L}0,i}(T_2)]}{\rho_{\text{L}0,i}(T_2)[\rho_{\text{B},i}(T_2) - \rho_{\text{L},i}(T_2)]} \right]$$

(28)

Porosity can be obtained from the relationship between pore ratio and porosity

$$n = \frac{e}{1+e} \times 100\%$$

(29)

Based on the measurement of $T_2$ energy spectrum area, the porosity and volume change rule of the shale specimen can be calculated, and the $T_2$ energy spectrum area after centrifugation at 300 psi can be calculated to be 1867 by Fig 13.

The link between the rock sample's volumetric expansion and porosity (or porosity ratio) can also be determined in simulated unloading disturbance testing, as shown in Eq. (30). The relationship curves of volume expansion and porosity with unloading amplitude can be established using the test data from Table 5, as illustrated in Fig 18.

**Table 5. Pore change of rock samples with unloading amplitudes.**

| Unloading amplitude/MPa <br> Calculation parameters | 0 | 10 | 20 | 30 | 40 | 50 | 60 |
|---|---|---|---|---|---|---|---|
| Saturated specimen mass/g | 64.151 | 64.156 | 63.887 | 63.916 | 64.054 | 64.142 | 63.989 |
| Specimen mass after centrifugation/g | 63.919 | 63.715 | 63.327 | 62.951 | 62.727 | 62.437 | 61.86 |
| Mass of discharged water/g | 0.232 | 0.441 | 0.56 | 0.965 | 1.327 | 1.705 | 2.129 |
| Discharge water volume/mm³ | 232 | 441 | 560 | 965 | 1327 | 1705 | 2129 |
| The ratio of pore space occupied by discharged water | 0.0095 | 0.0183 | 0.0234 | 0.0409 | 0.0572 | 0.0747 | 0.0950 |
| $T_2$ spectral area of the saturated specimen | 2417.7 | 3386.6 | 3785.4 | 5149.9 | 6373 | 7648.4 | 9078.5 |
| $T_2$ spectral area of the centrifuged specimen | 1896 | 1896 | 1896 | 1896 | 1896 | 1896 | 1896 |
| Porosity ratio | 0.0416 | 0.044 | 0.04804 | 0.0648 | 0.0814 | 0.0993 | 0.1201 |
| Porosity/% | 3.99 | 4.22 | 4.71 | 6.09 | 7.53 | 9.03 | 10.72 |
| Dilatation/% | 0 | 0.23 | 0.72 | 2.1 | 3.54 | 5.04 | 6.73 |

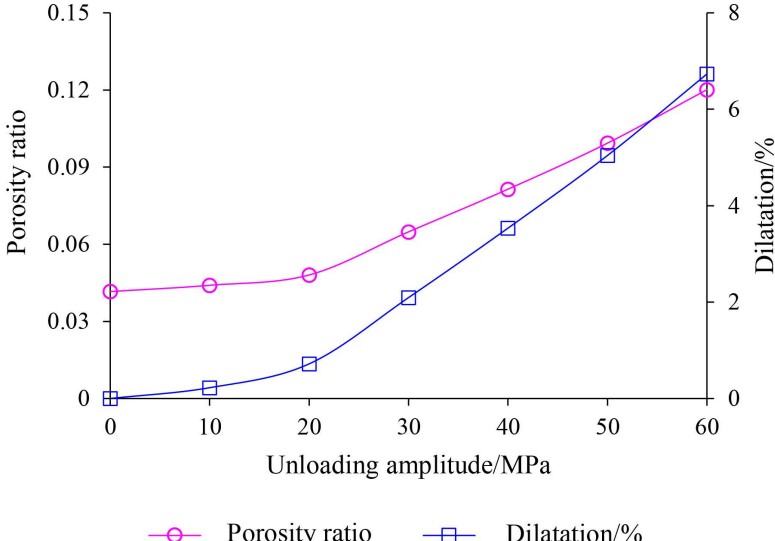

**Fig 18. Variation of shale void ratio and expansion rate after unloading disturbance.**

$$\eta_{\mathrm{u}} = \frac{e_{\mathrm{u}} - e_0}{1 + e_0} \times 100\% = n_{\mathrm{u}} - n_0$$

(30)

As can be seen from Fig 18, the pore ratio and swelling ratio of the shale specimens increased with increasing unloading amplitude and showed similar patterns. Specifically, the pore ratio and expansion rate of the specimen increase more slowly with the increase of unloading amplitude in small amplitude (less than 20 MPa); At substantial unloading (20MPa~60MPa), with the increase of unloading amplitude, it shows a significant tendency of increasing the pore ratio and expansion rate of the specimens. This result can provide theoretical guidance for stability control of rock bodies in deep underground engineering. In actual construction, a graded unloading strategy can be adopted for the highly stressed rock body: the unloading rate can be appropriately accelerated in the low-risk stage (unloading amplitude less than 20MPa); and when entering the high-risk stage (unloading amplitude of 20–60MPa), it is necessary to combine the anchor support, grouting reinforcement and other measures, and to strengthen the real-time monitoring, to avoid the risk of disasters, such as rock explosion, collapse and so on.

### 3.3 Analysis of mechanical properties of rock samples after unloading disturbance

The shale specimens under different unloading amplitudes were tested by triaxial compression test, and the test data were used to plot the stress-strain relationship curves, as shown in Fig 19.

The damage of rock specimens at different unloading amplitudes goes through four stages: Compressive closure stage, elastic linear deformation stage, initiation and extension stage and post-peak damage stage. The main deformation characteristics are as follows: (1) Before the point of destruction, the stress-strain curve basically shows a linear relationship; After the damage point, the stress-strain curve decreases rapidly. (2) With the increase of unloading amplitude, the compressive strength limit value of the rock has a significant decrease, and the modulus of elasticity also has a significant decrease, indicating that the unloading disturbance has a deterioration effect on the mechanical properties of the rock. As can be seen from Fig 20, with the increase of unloading amplitude, the shale shear strength limit value, decreases more slowly, which is consistent with the disturbance response of the fine-scale pore structure of the rock samples. The modulus of elasticity and shear strength also show a similar pattern of change, and then according to the intrinsic relationship between the modulus of elasticity and shear modulus, it is shown that the Poisson's ratio of shale does not change much.

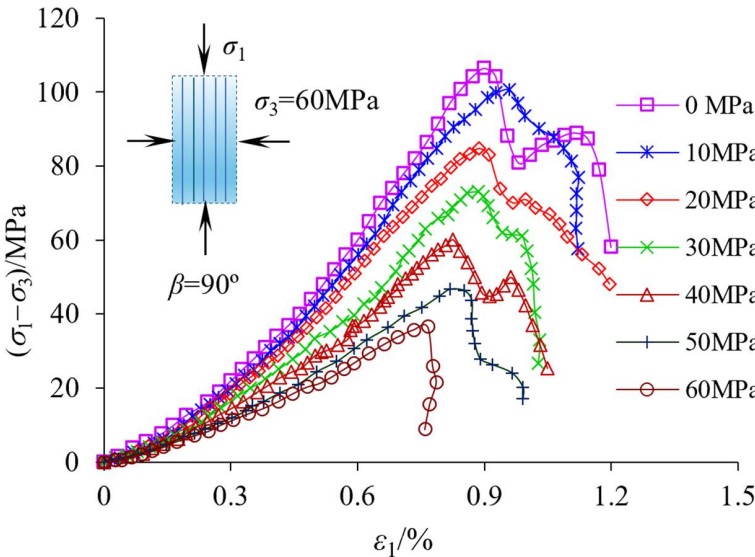

**Fig 19. Shale triaxial compression test after unloading disturbance.**

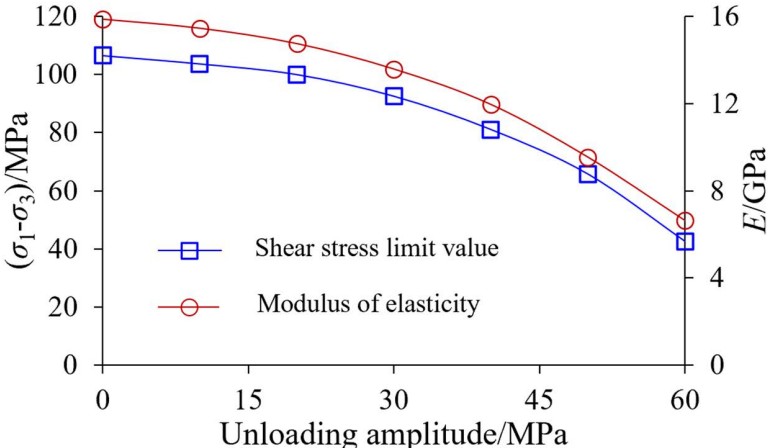

**Fig 20. Variation of shear strength and elastic modulus with different unloading amplitudes.**

## 4 Rock damage modeling

Working in underground spaces. Take the example of a well driller. Prior to the drilling of the borehole, the raw geopathic stresses are in a state of static equilibrium. The surrounding rock is under pressure while the borehole is being drilled. Meanwhile, before the mud cake is formed, the rock is permeable, and the drilling fluid does not act as a support. Changes in the stress state at various points in the well wall envelope cause the stress state to readjust. When the mud cake is formed, it can be used as an isolation layer due to its very low permeability and isolation of the drilling fluid from sex well wall percolation. In particular, when drilling through complex stratigraphic formations, the rock around the well is highly susceptible to induced cracking due to drilling unloading disturbances, which leads to a reduction in the strength of the rock, thus exacerbating the instability of the well wall. The application of the theory of elastic mechanics begins with

the following basic assumptions: the formation is complete and infinite, small deformation is assumed; the formation rocks are homogeneous and isotropic; the borehole is uniform, smooth, and has a good mud cake; and there are natural microfractures, cavities, and fragmentation in the wall of the well.

According to the theory of damage mechanics, taking the rock material strength before the disturbance as the initial state and the rock material strength after experiencing the unloading disturbance as the damage state, the basic equation of rock strength damage can be described as Eq. (31).

$$S_u = S_0 (1 - D_u)$$

(31)

Where: $S_0$ is the strength extremum of the rock material in the initial state, MPa; $S_u$ is the strength extremum of the rock material after the unloading disturbance, MPa; $D_u$ is the damage variable.

Weibull distribution can well analyze the microscopic damage of rock materials from the micro-metamorphic strength degradation law, and in this paper, the volume expansion rate is used as the examining variable, i.e., the damage probability density function $p(\eta)$ can be expressed as

$$p(\eta) = \frac{\beta}{F_0} \left( \frac{\eta}{F_0} \right)^{\beta-1} \exp \left[ -\left( \frac{\eta}{F_0} \right)^{\beta} \right], \ (\eta > 0)$$

(32)

Where: $\beta$ is the shape parameter, $F_0$ is the scale parameter, and $\eta$ is the expansion rate.

Referring to Krajcinovic et al. [57] who applied the combination of continuous damage mechanics theory and strength statistics theory to propose a statistical damage model, the author establishes some kind of factor Disturbance statistical damage model as shown in Eq. (33).

$$\sigma_i = E_0 \varepsilon_v \left( 1 - \frac{N_i}{N} \right)$$

(33)

Where: $E_0$ is the initial modulus of elasticity, MPa, $\varepsilon_v$ is the volumetric strain, %; $N_i$ is the number of microcells that fail under the effect of some Disturbance; $N$ is the total number of microcells.

Putting Eqs. (31) and (33) into perspective, one can define the random damage variable as the ratio of $N_i$ and $N$ and further assume that each disturbance produces some cells of damage of $N(\eta_i)$ when the volume expansion rate reaches a certain level $\eta_i$.

$$D_u = \frac{N(\eta_i)}{N} = \frac{\int_0^{\eta_i} NP(t)\, dt}{N}$$

(34)

Defects such as rock pores and cracks deep in the formation are generally tightly closed under the original state of ground stress, and there is some elastic deformation of the bedrock skeleton. To explore the change rule of the volume of rock in the original ground stress state, this paper through the triaxial compression test will return the perimeter pressure simulation to the original ground stress state (in this paper, the simulation of the perimeter rock imposed value of 60MPa), The volume change resulting from the unloading disturbance is then examined. The relationship between the volume change of the shale specimen under the state of ground stress and the volume change after pressure release can be deduced from the experimentally measured data, as shown in Eq. (35).

$$n'_u = \frac{V_u - \Delta V_c + \Delta V_s}{V - \Delta V_c} \times 100\%$$

(35)

Where: $n'_u$ is the porosity after returning to the formation pressure unloading disturbance, %, $V_u$ is the measured pore volume after unloading and disturbance, cm³, $V$ is the total volume of the rock specimen, cm³, $\Delta V_c$ is the change in rock specimen volume, cm³, for triaxial application of equivalent perimeter pressure, $\Delta V_s$ is the amount of change in bedrock volume, of the rock bedrock with equivalent perimeter pressure applied in three axes, cm³.

The amount of change in bedrock volume can be given by Eq. (36)

$$V_s = \frac{P_c(V - \Delta V_c)}{K} \tag{36}$$

Where: $P_c$ triaxial applied equivalent circumferential pressure value, MPa, $K$ is the rock bulk modulus, GPa.

There is a relationship between bulk modulus and modulus of elasticity and Poisson's ratio of rocks as shown in Eq. (37)

$$K = \frac{E}{3(1 - 2\mu)} \tag{37}$$

Substituting Eqs. (36) and (37) into (35) and simplifying yields the porosity after regression to formation pressure as Eq. (38)

$$n'_u = \left[ \frac{V_u - \Delta V_c}{V - \Delta V_c} + \frac{3P_c(1 - 2\mu)}{E} \right] \times 100\% \tag{38}$$

Then the rock expansion rate returning to the original stratigraphic stress state can also be obtained from the test data as

$$\eta'_u = \frac{V_u - \Delta V_c - V_f}{V - \Delta V_c} \times 100\% \tag{39}$$

Where: $\eta'_u$ is the volumetric expansion after regression to formation pressure, %, $V_f$ is the pore volume after regression to formation pressure, cm³.

Also, because of $V_f = n_0(V - \Delta V_c)$, substituting it into Eq. (39) yields

$$\eta'_u = \frac{V_u - \Delta V_c}{V - \Delta V_c} \times 100\% - n_0 \tag{40}$$

Where: $n_0$ original porosity of the formation, %.

The disturbance factor examined in this paper is the stress unloading disturbance. Substituting the unloading disturbance into the porosity and expansion expressions yields, as summarized by Table 6, The pore volume of the specimen after the unloading disturbance can be calculated, As well as after the unloading disturbance, triaxial tests were conducted to impose the statistics of the surrounding rock variables and to analyze the curves of volumetric expansion rate change under the unloading disturbance after shale regression to stratigraphic stress, as shown in Fig 21.

As can be seen from the figure, set $\eta'_u = aP_u^2 + bP_u + c$, where $a$, $b$, and $c$ are the experimental fit coefficients.

(1) Modeling of statistical damage

Substituting (31) into Eq. (34) yields

**Table 6. Porosity and expansion rate under unloading disturbance after returning to formation stress.**

| Unloading disturbance/MPa | 0 | 10 | 20 | 30 | 40 | 50 | 60 |
|---|---|---|---|---|---|---|---|
| $V$/cm³ | 24.579 | 24.581 | 24.478 | 24.489 | 24.542 | 24.575 | 24.517 |
| $V_u$/cm³ | 0.981 | 1.037 | 1.153 | 1.491 | 1.848 | 2.219 | 2.628 |
| $\Delta V_c$ /cm³ | 0.157 | 0.189 | 0.255 | 0.389 | 0.487 | 0.594 | 0.669 |
| $n'_u$ | 3.95 | 4.05 | 4.28 | 5.14 | 6.23 | 7.35 | 8.79 |
| $\eta'_u$ | 0 | 0.10 | 0.33 | 1.20 | 2.28 | 3.40 | 4.84 |

Note: $V_u$ is the measured pore volume after unloading disturbance, cm³, $n'_u$ and $\eta'_u$ are the porosity and volumetric expansion under stress relief disturbance after regression to stratigraphic stress, %.

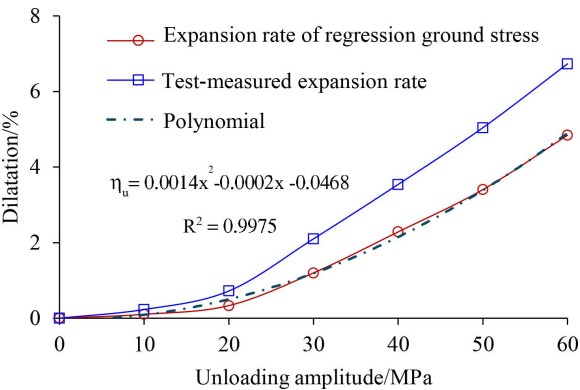

**Fig 21. Changing law of rock expansion rate with different unloading amplitude.**

$$D_u = 1 - \exp\left[-\left(\frac{aP_u^2 + bP_u + c}{F_{u0}}\right)^{\beta_u}\right] \tag{41}$$

Substituting Eqs. (41) into (33) yields a statistical damage evolution model for stress unloading disturbance, which describes the relationship between the rock strength $S_u$ and the initial strength $S_0$ and the unloading amplitude $P_u$ after the disturbance damage.

$$S_u = S_0 \exp\left[-\left(\frac{aP_u^2 + bP_u + c}{F_{u0}}\right)^{\beta_u}\right] \tag{42}$$

(2) Model parameterization

Eq. (42) can be obtained by simply shifting the terms and taking the logarithm of both sides of the equation after finishing:

$$-\ln\left(\frac{S_u}{S_0}\right) = \left(\frac{aP_u^2 + bP_u + c}{F_{u0}}\right)^{\beta_u} \tag{43}$$

 

The limiting values of rock shear strength and the damage variables calculated from Eq. (31) were obtained by triaxial compression test based on the stress unloading disturbance of the rock samples, as shown in Table 7.

By substituting the ultimate values of shear strength measured in the triaxial tests in Table 7 into Eq. (43), the values of the model parameters $F_{u0}$ and $\beta_u$ can be calculated, as shown in Table 8.

Substituting the model parameters from Table 8 into Eq. (41) yields the shale stress unloading disturbance statistical damage model function

$$D_u = 1 - \exp\left[-\left(\frac{aP_u^2 + bP_u + c}{3.1149}\right)^{-0.45775}\right]$$

(44)

A comparison of the damage test values calculated from the tests in Table 7 with the statistical damage model curves for the stress Unloading disturbance is shown in Fig 22.

As can be seen from Fig 22, the predicted curve of the model is consistent with the direction of the test curve, which meets the requirements of actual engineering design and research, and also provides the theoretical basis for the study of the rock mechanical deterioration behavior under the unloading disturbance of deep underground geotechnical engineering.

## 5 Conclusion

(1) The structural development and distribution characteristics of microscopic pores and cracks on the surface of the rock specimens were analyzed by scanning electron microscopy (SEM). To obtain accurate pore structure information, the

**Table 7. Rock shear strength limit value and damage variable after unloading disturbance.**

| Unloading amplitude/MPa | 0 | 10 | 20 | 30 | 40 | 50 | 60 |
|---|---|---|---|---|---|---|---|
| Shear strength limit value/MPa | 106.45 | 103.60 | 99.94 | 92.05 | 81.05 | 65.72 | 42.67 |
| Damage variable | 0 | 0.027 | 0.061 | 0.135 | 0.239 | 0.383 | 0.599 |

**Table 8. Model parameters.**

| Stress unloading disturbance | Model parameter | $\beta_u$ | $F_{u0}$ |
|---|---|---|---|
| | Parameter value | −0.45775 | 3.1149 |

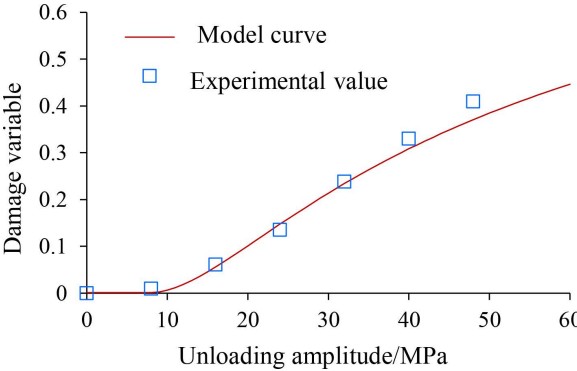

**Fig 22. Variation law of damage variables after unloading disturbance.**

SEM images were digitized using a combination of MATLAB image processing technology and image analysis technology (IPP6.0), and a method of quantifying damage using the $d_{max}/d_{min}$ ratio as a method of examining the development of micro- and micro-scale pore and crack evolution and expansion was proposed, which can quantify and analyze the size of the rock micro-porous structure porosity and pore ratio The method can quantitatively analyze the microporosity and pore ratio size of the rock.

(2) By analyzing the internal pore size and distribution characteristics of rock specimens through NMR technology, the relaxation time and pore size correspondence of the $T_2$ energy spectrum is used for pore classification, which also provides a quantitative analysis method for the expansion of pore evolution within rock specimens, i.e., the larger the $T_2$ amplitude is, the more the content of pore space corresponding to the size of that size is indicated; A smaller $T_2$ amplitude indicates a smaller content of pores corresponding to that size. Using the $T_2$ energy spectrum signal intensity test values, the $T_2$ energy spectrum area was calculated, and the rock porosity calculation formula was derived from the $T_2$ energy spectrum signal intensity test values.

(3) By using the stress control triaxial instrument to simulate the unloading disturbance of the surrounding rock of the well wall, and then using SEM and NMR testing techniques to analyze the change rule of the pore structure of the rock. It was found that with the increase of unloading amplitude, there was a significant increase in the number of nascent cracks with a tendency to expand, which was caused by shear extension; The internal microporous structure of the rock is relatively stable, and the medium and large pores show a trend of development and expansion and interconnection, indicating that the medium and large pores belong to the extended dominant pores.

(4) Triaxial compression test to analyze the change of shear strength and modulus of elasticity of rock after Unloading disturbance, it is found that with the increase of unloading amplitude, the ultimate value of shear strength and modulus of elasticity of rock decreases slowly, which is consistent with the Disturbance response of fine pore structure of the rock specimen, and combined with the theory of damage mechanics and strength, to establish a statistical damage model of the rock applicable to unloading of different amplitudes, and compared with the experimental values, and the model predictions are in good agreement with the experimental values. Compared with the test values, the predicted values of the model are in good agreement with the test, which is in line with the relevant practical engineering design requirements.

## Supporting information

**S1 File. The supporting documents contain detailed figure data.**
(DOCX)

## Author contributions

**Conceptualization:** Yanxu Liang, Haicheng She, Wangji Ding, Haidong She.

**Data curation:** Yanxu Liang.

**Funding acquisition:** Haicheng She.

**Methodology:** Yanxu Liang.

**Project administration:** Yanxu Liang.

**Writing – original draft:** Yanxu Liang.

**Writing – review & editing:** Haicheng She.

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
