## [Decision Letter · Decision Letter 0]

11 Jun 2025

PONE-D-25-26204Evolutionary Features of Microscopic Damage in Shale Under Unloading ActionPLOS ONE

Dear Dr. She,

Thank you for submitting your manuscript to PLOS ONE. After careful consideration, we feel that it has merit but does not fully meet PLOS ONE’s publication criteria as it currently stands. Therefore, we invite you to submit a revised version of the manuscript that addresses the points raised during the review process.

We look forward to receiving your revised manuscript.

Kind regards,

Kang Wang, Ph.D.

Academic Editor

PLOS ONE

4. We note that this submission includes NMR spectroscopy data. We would recommend that you include the following information in your methods section or as Supporting Information files:

a) The make/source of the NMR instrument used in your study, as well as the magnetic field strength. For each individual experiment, please also list: the nucleus being measured; the sample concentration; the solvent in which the sample is dissolved and if solvent signal suppression was used; the reference standard and the temperature.

b) A list of the chemical shifts for all compounds characterised by NMR spectroscopy, specifying, where relevant: the chemical shift (δ), the multiplicity and the coupling constants (in Hz), for the appropriate nuclei used for assignment.

c)The full integrated NMR spectrum, clearly labelled with the compound name and chemical structure.

We also strongly encourage authors to provide primary NMR data files, in particular for new compounds which have not been characterised in the existing literature. Authors should provide the acquisition data, FID files and processing parameters for each experiment, clearly labelled with the compound name and identifier, as well as a structure file for each provided dataset. See our list of recommended repositories here: https://journals.plos.org/plosone/s/recommended-repositories

“the Natural Science Foundation of Hubei Province (2024AFB878), the Key Laboratory of Reservoir and Dam Safety of the Ministry of Water Resources is open to research funds (YK323003) and Hubei Provincial Key Laboratory of Oil and Gas Drilling and Production Engineering Open Fund (YQZC202204)”

“This work was supported by the Natural Science Foundation of Hubei Province (2024AFB878), the Key Laboratory of Reservoir and Dam Safety of the Ministry of Water Resources is open to research funds (YK323003) and Hubei Provincial Key Laboratory of Oil and Gas Drilling and Production Engineering Open Fund (YQZC202204), which are gratefully acknowledged.”

“the Natural Science Foundation of Hubei Province (2024AFB878), the Key Laboratory of Reservoir and Dam Safety of the Ministry of Water Resources is open to research funds (YK323003) and Hubei Provincial Key Laboratory of Oil and Gas Drilling and Production Engineering Open Fund (YQZC202204)”

6. We note that your Data Availability Statement is currently as follows: [All relevant data are within the manuscript and its Supporting Information files.]

7. PLOS requires an ORCID iD for the corresponding author in Editorial Manager on papers submitted after December 6th, 2016. Please ensure that you have an ORCID iD and that it is validated in Editorial Manager. To do this, go to ‘Update my Information’ (in the upper left-hand corner of the main menu), and click on the Fetch/Validate link next to the ORCID field. This will take you to the ORCID site and allow you to create a new iD or authenticate a pre-existing iD in Editorial Manager.

Reviewers' comments:

Reviewer's Responses to Questions

**Comments to the Author**

1. Is the manuscript technically sound, and do the data support the conclusions?

Reviewer #1: Yes

Reviewer #2: Yes

Reviewer #3: Yes

Reviewer #4: Yes

2. Has the statistical analysis been performed appropriately and rigorously? 

Reviewer #1: Yes

Reviewer #2: Yes

Reviewer #3: Yes

Reviewer #4: Yes

3. Have the authors made all data underlying the findings in their manuscript fully available?

Reviewer #1: Yes

Reviewer #2: No

Reviewer #3: Yes

Reviewer #4: Yes

4. Is the manuscript presented in an intelligible fashion and written in standard English?

Reviewer #1: Yes

Reviewer #2: No

Reviewer #3: Yes

Reviewer #4: Yes

5. Review Comments to the Author

Reviewer #1: The manuscript addresses an important topic concerning the microscopic damage evolution in shale under unloading disturbances—a relevant issue in underground engineering applications such as drilling and mining. The experimental setup is well-designed, combining scanning electron microscopy (SEM), nuclear magnetic resonance (NMR), and triaxial tests to analyze damage characteristics and develop a statistical damage model. The integration of digital image processing for quantifying pore features and a mechanistic damage model is a commendable and novel approach.

However, the manuscript would benefit significantly from improvements in clarity, structure, and interpretation of results. Below is a detailed review:

1、 The manuscript contains numerous grammatical and syntactical issues that impede readability. Many sentences are fragmented or lack proper subject-verb agreement. Example from the abstract: “Through the design of rock unloading disturbance tests combined with electron microscope scanning…” is a sentence fragment and should be revised for clarity and grammatical correctness.

2、 The abstract should more clearly state the study’s objective, methodology, key findings, and significance. Currently, it is overly technical and lacks flow.

3、 While the introduction explains the significance of the study, it could benefit from more in-depth references to recent research in the past five years. Highlight gaps in the literature this study addresses, such as: compgeo.2024.106095; tafmec.2024.104691; fuel.2023.129584

4、 The methodology section (Section 2) is detailed but would benefit from being structured into subheadings like "Materials," "Test Protocol," and "Data Analysis Procedures."

5、 Figures and tables are well-detailed, but captions should be more informative. For instance, Fig. 8 should mention that different magnifications were used for different unloading amplitudes and why.

6、 The analysis of dmax/dmin and T2 spectrum shifts is insightful but should include statistical validation (e.g., ANOVA or regression) to support observed trends.

7、 Table 5 and Fig. 22 show clear trends in porosity and expansion, but these should be statistically evaluated to establish thresholds or nonlinear effects.

8、 The assumptions underlying the damage model (e.g., uniform initial stress state, isotropy) should be explicitly stated and discussed.

Reviewer #2: There are actually many studies on rock damage caused by unloading. The author conducted microscopic analysis using nuclear magnetic resonance and SEM. The research results have a certain degree of innovation, but there are mainly the following problems:

(1)The author's microscopic analysis was conducted after unloading, and the damage evolution characteristics during the unloading process cannot be obtained.

(2)SEM experiment is a destructive experiment. How does the author compare the effects of different unloading amplitude.

(3) Is the nuclear magnetic resonance experiment conducted using the same sample or different samples during the unloading experiment. If it is the same sample, is there interference between different unloading experiments? If it is different samples, is it still meaningful to compare them with each other.

(4)The author can refer to this paper "Strength weakening and its micromechanism in water–rock interaction, a short review in laboratory tests" on the advantages and disadvantages of micro experimental methods.

Reviewer #3: This manuscript analyzes the evolution laws of the pore types and mechanical properties on the surface and inside the rock sample after unloading disturbance. The evolution of pore cracks on the sample surface was studied through the classification of the dmax/dmin ratio. A damage quantification method was proposed, and a statistical damage ontology model of the unloading disturbed rock mass was constructed. However, after reviewing the paper carefully, there are still some details that need to modify. So, it is recommended that the paper should be modified significantly so as to increase its readability. Here are some suggestions for checking.

1. The variables that appear for the first time in the abstract need to be given complete words to facilitate readers' understanding. For example, "dmax/dmin".

2. The key words need to correspond to the title and abstract of the manuscript. However, the key words of this manuscript do not reflect the core characteristics of the research content.

3. The author proposes the influence of "unloading disturbances of different amplitudes" on the mechanical characteristics of rocks. What significance should be more clearly expounded in the introduction part in practical engineering?

4. The author describes the experimental equipment involved in the research process in Section 2.1 and lists the pictures of the relevant experimental instruments. In fact, listing these pictures is meaningless. The author should draw a flowchart to help readers understand the research ideas of the manuscript.

5. Section 3.1 of the manuscript describes the similarities in the properties of the selected samples. The first paragraph of section 3.2 presents the experimental scheme of the experimental study of rock unloading disturbance. It would be more reasonable to include these contents in the second chapter of the manuscript.

6. In Section 3.2 "(1) Scanning electron microscope image analysis", it is unreasonable for the author to select images with different magnifications as the original data for image digital processing. Furthermore, the author should elaborate on the basis that the selected scanning electron microscope images can represent the pore characteristics under different unloading amplitudes.

7. Section 3.3 is overly simplistic in analyzing the results of the triaxial force experiment and requires additional analysis content.

8. In Fig.23, the vertical coordinate shows “σ1-σ2”. Is it a true triaxial experiment or a miswrite?

Reviewer #4: This paper explores the micro-damage evolution characteristics and mechanical property changes of shale under unloading through experiments and theoretical analysis, and establishes a damage constitutive model. The research content is rich and has certain engineering value, but there are still some deficiencies in the writing. Therefore, I suggest a major revision. The following are the revision suggestions:

1. The author's statement that "research on rock unloading damage is relatively scarce" is questionable. Researchers have achieved abundant results in this field. Therefore, I suggest that the author reorganize the literature.

2. The introduction and result analysis and discussion are not in-depth enough, lacking in-depth analysis based on fracture mechanics. The following literature is provided for the author's reference:

Analytical solution of the stress field and plastic zone at the tip of a closed crack. Frontiers in Earth Science. 2024.

3. In lines 168-172, why are different magnifications set? Please provide additional explanations in the text. Moreover, inconsistent magnifications of the images may lead to inconsistent results in pore statistics. It is suggested to unify the magnification or explain the comparability of the data under different magnifications.

4. In line 231, the gray-scale threshold is set to 50 (Formula 4), but the basis for selecting this threshold is not explained. It is suggested to verify the rationality of the threshold through sensitivity analysis or to supplement relevant references to validate its rationality.

5. Figure 25 needs to be redrawn to ensure the clarity of the image and the uniformity of the font format.

6. It is suggested to add a chapter on "Research Limitations and Prospects", to further explain the limitations of the model (whether it is applicable to other rocks) and the limitations of the experiments (the selection of unloading rate and amplitude).

6. PLOS authors have the option to publish the peer review history of their article (what does this mean? ). If published, this will include your full peer review and any attached files.

**Do you want your identity to be public for this peer review?** For information about this choice, including consent withdrawal, please see our Privacy Policy .

Reviewer #1: No

Reviewer #2: No

Reviewer #3: **Yes: ** Qiangui Zhang

Reviewer #4: No

---

## [Author Response · Author response to Decision Letter 1]

9 Jul 2025

Dear Dr. Wang,

Thank you for giving us the opportunity to submit a revised draft of the manuscript “Evolutionary Features of Microscopic Damage in Shale Under Unloading Action” for publication in the Journal of PLOS ONE. We appreciate the time and effort that you and the reviewers dedicated to providing feedback on our manuscript and are grateful for the insightful comments on and valuable improvements to our paper. We have incorporated the suggestions made by the reviewers. Those changes are highlighted within the manuscript. Please see below, in blue, for a point-by-point response to the reviewers’ comments and concerns.

Reviewers' Comments to the Authors:

Reviewer 1

The manuscript addresses an important topic concerning the microscopic damage evolution in shale under unloading disturbances—a relevant issue in underground engineering applications such as drilling and mining. The experimental setup is well-designed, combining scanning electron microscopy (SEM), nuclear magnetic resonance (NMR), and triaxial tests to analyze damage characteristics and develop a statistical damage model. The integration of digital image processing for quantifying pore features and a mechanistic damage model is a commendable and novel approach.

However, the manuscript would benefit significantly from improvements in clarity, structure, and interpretation of results. Below is a detailed review:

Author response: We feel great thanks for your professional review work on our article. As you are concerned, several problems need to be addressed. According to your nice suggestions, we have made extensive corrections to our previous draft; the detailed corrections are listed below.

1、 The manuscript contains numerous grammatical and syntactical issues that impede readability. Many sentences are fragmented or lack proper subject-verb agreement. Example from the abstract: “Through the design of rock unloading disturbance tests combined with electron microscope scanning…” is a sentence fragment and should be revised for clarity and grammatical correctness.

Author response: We are very grateful to the reviewers for their scrutiny. We have rewritten the abstract section based on the reviewers' suggestions. The details are as follows:

To reveal the microscopic damage evolution law of rocks under the effect of unloading disturbances with different amplitudes, electron microscope scanning, nuclear magnetic resonance (NMR), and triaxial compression tests were carried out. The evolution patterns of surface and internal pore types and mechanical properties of rock specimens after unloading perturbation were analyzed. In this paper, a classification of the ratio of dmax/dmin (dmax and dmin refer to the maximum and minimum pore size of each pore, respectively) is proposed to examine the pore and crack evolution extension development on the surface of the specimen. Meanwhile, the T2 energy spectrum with pore size classification is used to examine the damage quantification law of the pore and crack extension evolution inside the specimen. Finally, the statistical damage model of unloaded disturbed rock is established through theoretical derivation, and the accuracy of the model is verified by experimental data. The study shows that: (1) With the increase of unloading amplitude, there is an increase in the number of nascent cracks and a tendency to expand, which is caused by shear extension cracks; with the increase of unloading amplitude, there is a tendency for the microporosity to shift to the mesoporosity, and the mesoporosity has a tendency to shift to the macroporosity, and there is a decrease in the number of micropores as a whole, which indicates that there is almost no new pore sprouting. (2) When the unloading amplitude is less than 20MPa, with the increase of the unloading amplitude, the pore ratio and expansion rate of the specimen increase slowly; when the unloading amplitude is more than 20MPa, with the increase of the unloading amplitude, the pore ratio and expansion rate of the specimen have a significant tendency to increase. (3) With the increase of unloading amplitude, the shale shear strength limit value decreases more slowly, the modulus of elasticity and shear strength also show a similar pattern of change, and the same way to derive the rock Poisson's ratio does not change much.

We've highlighted the changes in yellow in the revised version. Please review the revised draft for specific changes.

2、 The abstract should more clearly state the study’s objective, methodology, key findings, and significance. Currently, it is overly technical and lacks flow.

Author response: We agree with the reviewer’s assessment. We have reorganized the abstract section. Modifications as above.

We've highlighted the changes in yellow in the revised version. Please review the revised draft for specific changes.

3、 While the introduction explains the significance of the study, it could benefit from more in-depth references to recent research in the past five years. Highlight gaps in the literature this study addresses, such as: compgeo.2024.106095; tafmec.2024.104691; fuel.2023.129584

Author response: We are grateful to the reviewers for their careful review of the manuscript. We researched the most recent literature in more depth. We focused on the literature recommended by the reviewers. This literature has been very enlightening. We have added these relevant references to the manuscript. The details are as follows:

With the increasing number of tunnels, pits, slopes, and other projects, the problem of disturbance by various factors has become more and more prominent [1–3]. In these works, the soil and rock bodies are inevitably subjected to unloading disturbances of varying magnitudes, which in turn produce a range of disturbance effects [4]. These disturbance effects will not only affect the stability and safety of the project but also have a certain impact on the surrounding environment [5]. Therefore, the study of the damage characteristics of rocks under the action of unloading perturbation is an important scientific guidance for safety, stability, economy and long-term service performance in engineering.

In recent years, many scholars have done a lot of research on the mechanical properties of rocks under the action of various factors of perturbation, etc., and have achieved rich research results. In terms of impact perturbation: taking red sandstone as the research object, the impact disturbance test was conducted on red sandstone to analyze the mechanical response and deformation mechanism of red sandstone after being subjected to impact disturbance [6], which revealed the damage evolution, critical damage amount, crack extension law and damage mode of red sandstone under repeated impact loads [7]. In terms of unloading disturbance: Fu et al. [8] designed three unloading circumferential pressure tests of sandstone under different initial axial pressures, analyzed the change characteristics of rock porosity and T2 spectral curves by NMR technique, and established the relationship between the degree of damage and the unloading circumferential pressure ratio. Zhou et al. [9] conducted unloaded perimeter pressure tests on rock specimens and NMR tests on unloaded specimens to study the fine damage evolution characteristics of unloaded rocks to obtain the change rules of stress-strain curves, rock porosity, and NMR parameters during the damage process. Shi et al. [10] used true triaxial unloading perturbation tests. Mechanical properties of rocks under different stresses were studied. The crack evolution laws of rocks in different environments were also analyzed. Guo et al. [11] characterized the deformation and fracture patterns of shale samples under different stress paths by triaxial unloading tests under different stress paths. Lei et al. [12] conducted a series of loading and unloading tests on the rock mass around the roadway using high-precision acoustic emission technology. The results of the study provide guidance for deep shaft tunnel support work and disaster prevention and control. Combined with specific engineering examples, the mechanical property changes of soft rock in the unloading process were studied. [13] The effects of unloading rate, initial stress state, and other factors on the mechanical properties of soft rock were analyzed. [14] The deformation, damage mode, and strain energy evolution were studied in conjunction with discrete element simulation, and it was found that the faster the unloading rate, the more violent the damage was. [15] The unloading rate of the soft rock was also analyzed. In terms of hydration disturbance: Pu et al. [16] used SHPB experiments to study the dynamic compressive strength change of sandstone after wet and dry cycles, and used ultrasonic waves to detect the change of wave velocity of sandstone before and after wet and dry cycles, which reflected the change rule of the internal structure of the rock; Dong et al. [17] carried out a uniaxial compression test on sandstone samples with different submergence heights, and used a high-speed camera and an acoustic emission monitoring system to simultaneously monitoring the damage process, and established a damage evolution model using damage theory. Zhang et al [18] analyzed the effects of shale hydration from macro and micro perspectives through compression tests and CT scan tests. In terms of temperature disturbance: Park et al. [19] conducted freeze-thaw cycling tests on rock specimens and used X-ray computed tomography (CT) and scanning electron microscopy (SEM) to obtain images of microstructural changes within the rock, as well as to measure the changes in the physical properties, and investigated the effects of freeze-thawing of water within the pore spaces, cracks, and seams of the rock on the microstructure and physical properties of the rock. Wang et al. [20] conducted a detailed investigation of the mechanical behavior of rocks after temperature disturbance. In terms of multifactorial perturbations: Ma et al. [21] developed a mathematical model to analyze the stability of the wellbore for the stresses caused by mechanical, hydraulic, and chemical actions, and analyzed the causes of collapse in shale formations. Ekbote and Abousleiman [22] developed a multi-field coupled stress field model for the well wall, but it was not directly applied to evaluate the stability of the well wall. It has also been investigated based on fracture mechanics how factors such as crack angle, circumferential pressure and material properties affect the stress field, displacement field, plastic zone size and crack extension direction [23].

In this regard, scholars have used various methods to carry out disturbance experiments on rock specimens, which can be categorized into physical simulation tests, numerical simulation tests, and derivation of theories. There are many ways of physical simulation tests, such as triaxial compression test [24–27], uniaxial compression test [28,29], power impact test [30–32], CO2 fracturing tests [33,34], etc. The numerical simulation test is a more mainstream test means [35–41]. It is also possible to combine experimental and numerical simulations, and then determine the accuracy of the experimental simulations through theoretical derivation [42–45]. However, post-test detection methods are less innovative, usually acoustic emission [46–49], CT scanning technology [50,51], nuclear magnetic resonance (NMR) techniques [52], and electron microscope scanning techniques [53] are used, The results of the tests and inspections were not analyzed using digital processing.

In summary, many scholars have achieved rich results in the research related to the disturbance of rocks by various factors. However, fewer test results have been digitized. In this paper, the deep shale is studied, and first, their composition and surface microstructure were examined; Second, a stress-controlled triaxial instrument was used to simulate the unloading disturbance effect on the rock; Then, the variation rules of microstructural and mechanical properties of rocks under unloading conditions of different magnitudes were analyzed by SEM and NMR; Finally, damage mechanics and statistical strength theory are combined to model the statistical damage variables of rock unloading disturbances.

18. Zhang Q, Wang L, Zhao P, Fan X, Zeng F, Yao B, et al. Mechanical Properties of Lamellar Shale Considering the Effect of Rock Structure and Hydration from Macroscopic and Microscopic Points of View. Applied Sciences. 2022;12: 1026. doi:10.3390/app12031026

23. Wu G, Wang W, Peng S. Analytical solution of the stress field and plastic zone at the tip of a closed crack. Front Earth Sci. 2024;12: 1370672. doi:10.3389/feart.2024.1370672

33. Wang K, Pan H, Fujii Y. Study on energy distribution and attenuation of CO2 fracturing vibration from coal-like material in a new test platform. Fuel. 2024;356: 129584. doi:10.1016/j.fuel.2023.129584

34. Wang K, Chang C. Study on the characteristics of CO2 fracturing rock damage based on fractal theory. Theor Appl Fract Mech. 2024;134: 104691. doi:10.1016/j.tafmec.2024.104691

40. Li ZX, Fujii Y, Alam AKMB, Li ZH, Du F, Wei WJ. Implementing a simple 2D constitutive model for rocks into finite element method. Computers and Geotechnics. 2024;167: 106095. doi:10.1016/j.compgeo.2024.106095

We've highlighted the changes in yellow in the revised version. Please review the revised draft for specific changes.

4、 The methodology section (Section 2) is detailed but would benefit from being structured into subheadings like "Materials," "Test Protocol," and "Data Analysis Procedures."

Author response: We are very grateful to the reviewers for their scrutiny. We agree with the reviewer’s assessment. We have reorganized the content and title of the article based on your suggestions with the other reviewers. The specific titles are listed below:

1 Introduction

2 Experimental program

2.1 Experimental instruments and processes

2.2 Experimental study of the components and microstructure of the specimen

2.3 Experimental design of rock unloading disturbance

3 Experimental studies

3.1 Analysis of Scanning electron microscope image

3.2 Analysis of NMR test results

3.3 Analysis of mechanical properties of rock samples after unloading disturbance

4 Rock damage modeling

5 Conclusion

We've highlighted the changes in yellow in the revised version. Please review the revised draft for specific changes.

5、 Figures and tables are well-detailed, but captions should be more informative. For instance, Fig. 8 should mention that different magnifications were used for different unloading amplitudes and why.

Author response: We are very grateful to the reviewers for their scrutiny. We analyzed the surface microstructure of rocks at different unloading amplitudes by electron microscope scanning tests. We performed scanning tests at different multiples for each group of rocks at different unloading amplitudes. However, we believe that the selection of images with different magnifications does not affect the results of the pore statistics. We focused on the maximum and minimum areas for the SEM image parameter statistics. We believe that the comparison of microstructures of shale specimens is reasonable when the range of maximum and minimum areas is controlled from 13 to 10095 �m2. At the same time, we excluded from the study graphs (a) that did not fit into the control scope category. And, we are comparing by the ratio of pore types on a local scale. We apologize for the possible lack of clarity in the presentation of the manuscript. We have included appropriate explanations in the manuscript. The details are as follows:

According to Table 2's statistical results, the minimum area in the microfine view pore parameters of the rock varies in the statistics because of the various SEM magnifications. For example, Fig. 6a shows that at 90 times magnification, the minimum area of the statistics is 115 �m2, while Figs. 6b to f shows that at 1416~ 3000 times magnification, the minimum area of the statistics is 10~13 �m2. Similarly, the existence of significant lysimeters or voids in the scanned region causes variations in the maximum area measured. Controlling the pore size (pore area) in

---

## [Decision Letter · Decision Letter 1]

17 Jul 2025

Evolutionary Features of Microscopic Damage in Shale Under Unloading Action

PONE-D-25-26204R1

Dear Dr. She,

We’re pleased to inform you that your manuscript has been judged scientifically suitable for publication and will be formally accepted for publication once it meets all outstanding technical requirements.

Kind regards,

Kang Wang, Ph.D.

Academic Editor

PLOS ONE

Additional Editor Comments (optional):

Reviewers' comments:

Reviewer's Responses to Questions

**Comments to the Author**

1. If the authors have adequately addressed your comments raised in a previous round of review and you feel that this manuscript is now acceptable for publication, you may indicate that here to bypass the “Comments to the Author” section, enter your conflict of interest statement in the “Confidential to Editor” section, and submit your "Accept" recommendation.

Reviewer #1: (No Response)

Reviewer #2: All comments have been addressed

Reviewer #3: All comments have been addressed

Reviewer #4: All comments have been addressed

2. Is the manuscript technically sound, and do the data support the conclusions?

Reviewer #1: (No Response)

Reviewer #2: Yes

Reviewer #3: Yes

Reviewer #4: Yes

3. Has the statistical analysis been performed appropriately and rigorously? 

Reviewer #1: (No Response)

Reviewer #2: Yes

Reviewer #3: Yes

Reviewer #4: Yes

4. Have the authors made all data underlying the findings in their manuscript fully available?

Reviewer #1: (No Response)

Reviewer #2: Yes

Reviewer #3: Yes

Reviewer #4: Yes

5. Is the manuscript presented in an intelligible fashion and written in standard English?

Reviewer #1: (No Response)

Reviewer #2: Yes

Reviewer #3: Yes

Reviewer #4: Yes

6. Review Comments to the Author

Reviewer #1: (No Response)

Reviewer #2: The author has made detailed revisions based on the suggestions and comments, and I believe this paper can be publish ed

Reviewer #3: All the comments have been addressed in the new manuscript, and all the questions have been well answered. I suggest that it should be published.

Reviewer #4: The author has carefully revised the paper based on the opinions of the reviewers. I think the paper can already be accepted

7. PLOS authors have the option to publish the peer review history of their article (what does this mean? ). If published, this will include your full peer review and any attached files.

**Do you want your identity to be public for this peer review?** For information about this choice, including consent withdrawal, please see our Privacy Policy .

Reviewer #1: No

Reviewer #2: No

Reviewer #3: No

Reviewer #4: No

---

## [Editor Report · Acceptance letter]

PONE-D-25-26204R1

PLOS ONE

Dear Dr. She,

I'm pleased to inform you that your manuscript has been deemed suitable for publication in PLOS ONE. Congratulations! Your manuscript is now being handed over to our production team.

Kind regards,

on behalf of

Dr. Kang Wang

Academic Editor

PLOS ONE